# Monocular Visual SLAM Based on a Cooperative UAV–Target System

**DOI:** 10.3390/s20123531

**Published:** 2020-06-22

**Authors:** Juan-Carlos Trujillo, Rodrigo Munguia, Sarquis Urzua, Edmundo Guerra, Antoni Grau

**Affiliations:** 1Department of Computer Science, CUCEI, University of Guadalajara, Guadalajara 44430, Mexico; juancarlos_max@hotmail.com (J.-C.T.); isi.sarquis@gmail.com (S.U.); 2Department of Automatic Control, Technical University of Catalonia UPC, 08034 Barcelona, Spain; edmundo.guerra@upc.edu (E.G.); antoni.grau@upc.edu (A.G.)

**Keywords:** state estimation, unmanned aerial vehicle, monocular SLAM, observability, cooperative target, flight formation control

## Abstract

To obtain autonomy in applications that involve Unmanned Aerial Vehicles (UAVs), the capacity of self-location and perception of the operational environment is a fundamental requirement. To this effect, GPS represents the typical solution for determining the position of a UAV operating in outdoor and open environments. On the other hand, GPS cannot be a reliable solution for a different kind of environments like cluttered and indoor ones. In this scenario, a good alternative is represented by the monocular SLAM (Simultaneous Localization and Mapping) methods. A monocular SLAM system allows a UAV to operate in a priori unknown environment using an onboard camera to simultaneously build a map of its surroundings while at the same time locates itself respect to this map. So, given the problem of an aerial robot that must follow a free-moving cooperative target in a GPS denied environment, this work presents a monocular-based SLAM approach for cooperative UAV–Target systems that addresses the state estimation problem of (i) the UAV position and velocity, (ii) the target position and velocity, (iii) the landmarks positions (map). The proposed monocular SLAM system incorporates altitude measurements obtained from an altimeter. In this case, an observability analysis is carried out to show that the observability properties of the system are improved by incorporating altitude measurements. Furthermore, a novel technique to estimate the approximate depth of the new visual landmarks is proposed, which takes advantage of the cooperative target. Additionally, a control system is proposed for maintaining a stable flight formation of the UAV with respect to the target. In this case, the stability of control laws is proved using the Lyapunov theory. The experimental results obtained from real data as well as the results obtained from computer simulations show that the proposed scheme can provide good performance.

## 1. Introduction

Nowadays, unmanned aerial vehicles (UAVs), computer vision techniques, and flight control systems have received great attention from the research community in robotics. This interest has resulted in the development of systems with a high degree of autonomy. UAVs are very versatile platforms and very useful for several tasks and applications [1,2]. In this context, a fundamental problem to solve is the estimation of the positions of UAVs. For most applications, GPS (Global Positioning System) still represents the main alternative for addressing the localization problem of UAVs. However, GPS comes with some well-known drawbacks associated with its use. For instance, in scenarios where GPS signals are jammed intentionally [3] or when the precision error can be substantial and they provide poor operability due to multipath propagation (e.g., natural and urban canyons [4,5]). Furthermore, there are scenarios where the GPS is inaccessible (e.g., indoor). Hence, to improve accuracy and robustness, additional sensory information, like visual data, can be integrated into the system. Cameras are lightweight, inexpensive, power-saving, and provide lots of information, moreover, they are well adapted to be integrated into embedded systems. In this context, visual SLAM methods are important options that allow a UAV to operate in an a priori unknown environment using only on-board sensors to simultaneously build a map of its surroundings while, at the same time, locating itself in respect to this map. On the other hand, perhaps the most important challenge associated with the application of monocular SLAM techniques has to do with the metric scale [6]. In monocular SLAM systems, the metric scale of the scene is difficult to retrieve, and even if the metric scale is known as an initial condition, the metric scale tends to degenerate (drift) if the system does not incorporate continuous metric information.

Many works can be found in the literature where visual-based SLAM methods are used for UAV navigation tasks (e.g., [7,8]). For SLAM based on monocular vision, different approaches have been followed for addressing the problem of the metric scale. In [9], the position of the first map features is determined by knowing the metrics of an initial pattern. In [10], a method with several assumptions about the structure of the environment is proposed; one of these assumptions is the flatness of the floor. This restricts the use of the method to very specific environments. Other methods like [11] or [12] fuse inertial measurements obtained from an inertial measurement unit (IMU) to recover the metric scale. A drawback associated with this approach has to do with the dynamic bias of the accelerometers which is very difficult to estimate. In [13], the information given by an altimeter is added to the monocular SLAM system to recover the metric scale.

The idea of applying cooperative approaches of SLAM to UAVs has also been explored. For example, [14,15] present a Kalman-filter-based centralized architecture. In [16,17,18], monocular SLAM methods for cooperative multi-UAV systems are presented to improve navigation capabilities in GPS-challenging environments. In [19], the idea of combining monocular SLAM with cooperative human–robot information to improve localization capabilities is presented. Furthermore, a single-robot SLAM approach is presented in [20], where the system state is augmented with the state of the dynamic target. In that work, robot position, map, and target are estimated using a Constrained Local Submap Filter (CLSF) based on an Extended Kalman filter (EKF) configuration. In [21], the problem of cooperative localization and target tracking with a team of moving robots is addressed. In this case, a least-squares minimization approach is followed and solved using sparse optimization. However, the main drawback of this method is related to the fact that the positions of landmarks are assumed a priori. In [22], a range-based cooperative localization method is proposed for multiple platforms with different structures. In this case, the dead reckoning system is implemented by means of an adaptive ant colony optimization particle filter algorithm. Furthermore, a method that incorporates the ultra-wideband technology into SLAM is presented in [23].

In a previous work by the authors [24], the problem of cooperative visual-SLAM based tracking of a lead agent was addressed. With big differences from the present work, where the (single robot) monocular-SLAM problem is addressed, in [24] a team of aerial robots in flight formation had to follow the dynamic lead agent. When two or more camera-robots are considered in the system, the problem of landmark initialization, as well as the problem of recovering the metric scale of the world, can be solved using a visual pseudo-stereo approach. On the other hand, the former problems can constitute a bigger challenge, if only a camera-robot is available in the system. This work deals with this last scenario.

### 1.1. Objectives and Contributions

Recently, in [25], a visual SLAM method using an RGB-D camera was presented. In that work, the information given by the RGB-D camera is used to directly obtain depth information of its surroundings. However, the depth range of that kind of camera is quite limited. In [26], a method for the initialization of characteristics in visual SLAM, employing the algorithm based on planar homography constraints, is presented. In that case, it is assumed that the camera only moves in a planar scene. In [27], a visual SLAM system that integrates a monocular camera and a 1D-laser range finder is presented; it seeks to provide scale recovering and drift correction. On the other hand, LiDAR-like sensors are generally expensive and can over weigh the system for certain applications presenting moving parts which can induce some errors.

Trying to present an alternative to related approaches, in this work, the use of a visual-based SLAM scheme is studied for addressing the problem of estimating the position of an aerial robot and a cooperative target in GPS-denied environments. The general idea is to use a set of a priori unknown static natural landmarks and the cooperation between a UAV and a target for locating both the aerial robot and the target moving freely in the 3D space. This objective is achieved using (i) monocular measurements of the target and the landmarks, (ii) measurements of altitude of the UAV, and (iii) range measurements between UAV and target.

The well-known EKF-SLAM methodology is used as the main estimation technique for the proposed cooperative monocular-based SLAM scheme. In this work, since the observability plays a key role in the convergence and robustness of the EKF ([28,29]), the observability properties of the system are analyzed using a nonlinear observability test. In particular, it is shown that by the sole addition of altitude measurement, the observability properties of the SLAM system are improved. In this case, the inclusion of the altimeter in monocular SLAM has been proposed previously in other works, but no such observability analyses have been done before.

In monocular-based SLAM systems, the process of initializing the new landmarks into the system state plays an important role in the performance of the system as well [30]. When only monocular measurements of landmarks are available, it is not easy to obtain 3D information from them. In this case, it becomes a difficult task to properly initialize the new map features into the system state due to the missing information. Therefore, a novel technique to estimate the approximate depth of the new visual landmarks is proposed in this work. The main idea is to take advantage of the UAV–Target cooperative scheme to infer the depth of landmarks near the target. In this case, it is shown that by the addition of altitude measurements and by the use of the proposed initialization technique, the problem of recovering the metric scale is overcome.

This work also presents a formation control scheme that allows carrying out the formation of the UAV with respect to the target. Moreover, the stability of the control system is assured utilizing the Lyapunov theory. In simulations, the state estimated by the SLAM system is used as a feedback to the proposed control scheme to test the closed-loop performance of both the estimator and the control. Finally, experiments with real data are presented to validate the applicability and performance of the proposed method.

### 1.2. Paper Outline

This work presents the following structure: mathematical models and system specifications are presented in Section 2. The nonlinear observability analysis is presented in Section 3. The proposed SLAM approach is described in Section 4. The control system is described in Section 5. Section 6 shows the results obtained from numerical simulations and with real data experiments. Finally, conclusions and final remarks of this work are given in Section 7.

## 2. System Specification

In this section, the mathematical models that will be used in this work are introduced. First, the model used for representing the dynamics of a UAV–camera system, and the model used for representing the dynamics of the target are described. Then, the model for representing the landmarks as map features is described. Furthermore, measurement models are introduced: (i) the camera projection model, (ii) the altimeter measurement model, and (iii) the range measurement model.

In applications like aerial vehicles, the attitude and heading (roll, pitch, and yaw) estimation is properly handled with available AHRS systems (e.g., [31,32]), so in this work, the estimated attitude of the vehicle is assumed to be provided by an Attitude and Heading Reference Systems (AHRS) as well as the orientation of the camera pointing always toward the ground. In practice, the foregoing assumption can be easily addressed, for instance, with the use of a servo-controlled camera gimbal or digital image stabilization (e.g., [33]). To this effect, it is important to note that the use of reliable commercial-degree AHRS and gimbal devices are assumed.

Taking into account the previous considerations, the system state can be simplified by removing the variables related to attitude and heading (which are provided by the AHRS). Therefore, the problem will be focused on the position estimation.

### 2.1. Dynamics of the System

Let consider the following continuous-time model describing the dynamics of the proposed system (see Figure 1): (1)$$\dot{x}=[\begin{array}{c} {\dot{x}}_{t}\\ {\dot{v}}_{t}\\ {\dot{x}}_{c}\\ {\dot{v}}_{c}\\ {\dot{x}}_{a}^{i} \end{array}]=[\begin{array}{c} \;\;v_{t}\\ 0_{3\times 1}\\ \;\;v_{c}\\ 0_{3\times 1}\\ 0_{3\times 1} \end{array}]$$
where the state vector x is defined as:(2)$$x={\left[\begin{array}{ccccc} x_{t} & v_{t} & x_{c} & v_{c} & x_{a}^{i} \end{array}\right]}^{T}$$
with $i=1,\ldots ,n_{1}$, where $n_{1}$ is the number of landmarks included into the map. In this work, the term *landmarks* will be used to refer to natural features of the environment that are detected and tracked from the images acquired by a camera.

Additionally, let $x_{t}={\left[\begin{array}{ccc} x_{t} & y_{t} & z_{t} \end{array}\right]}^{T}$ represent the position (in meters) of the target, with respect to the reference system *W*. Let $x_{c}={\left[\begin{array}{ccc} x_{c} & y_{c} & z_{c} \end{array}\right]}^{T}$ represent the position (in meters) of the reference system *C* of the camera, with respect to the reference system *W*. Let $v_{t}={\left[\begin{array}{ccc} {\dot{x}}_{t} & {\dot{y}}_{t} & {\dot{z}}_{t} \end{array}\right]}^{T}$ represent the linear velocity (in $\frac{m}{s}$) of the target. Let $v_{c}={\left[\begin{array}{ccc} {\dot{x}}_{c} & {\dot{y}}_{c} & {\dot{z}}_{c} \end{array}\right]}^{T}$ represent the linear velocity (in $\frac{m}{s}$) of the camera. Finally, let $x_{a}^{i}={\left[\begin{array}{ccc} x_{a}^{i} & y_{a}^{i} & z_{a}^{i} \end{array}\right]}^{T}$ be the position of the *i-th* landmark (in meters) with respect to the reference system *W*. In Equation (Equation 1), the UAV–camera system, as well as the target, is assumed to move freely in the three-dimensional space. Let note that a non-acceleration model is assumed for the UAV–camera system and the target. Non-acceleration models are commonly used in monocular SLAM systems. In this case, it will be seen in Section 4 that unmodeled dynamics are represented by means of zero-mean Gaussian noise. In any case, augmenting the target model to consider higher-order dynamics could be straightforward. Furthermore, note that landmarks are assumed to remain static.

### 2.2. Camera Measurement Model for the Projection of Landmarks

Let consider the projection of a single landmark over the image plane of a camera. Using the pinhole model [34] the following expression can be defined:(3)$$\begin{array}{c} h_{c}^{i}=[\begin{array}{c} u_{c}^{i}\\ v_{c}^{i} \end{array}]=\frac{1}{z_{d}^{i}}[\begin{array}{cc} \frac{f_{c}}{d_{u}} & 0\\ 0 & \frac{f_{c}}{d_{v}} \end{array}][\begin{array}{c} x_{d}^{i}\\ y_{d}^{i} \end{array}]+[\begin{array}{c} c_{u}+d_{ur}+d_{ut}\\ c_{v}+d_{vr}+d_{vt} \end{array}] \end{array}$$

Let $[u_{c}^{i}$,$v_{c}^{i}]$ define the coordinates (in pixels) of the projection of the *i*-th landmark over the image of the camera. Let $f_{c}$ be the focal length (in meters) of the camera. Let $\left[d_{u},d_{v}\right]$ be the conversion parameters (in m/pixel) for the camera. Let $\left[c_{u},c_{v}\right]$ be the coordinates (in pixels) of the image central point of the camera. Let $\left[d_{ur},d_{vr}\right]$ be components (in pixels) accounting for the radial distortion of the camera. Let $\left[d_{ut},d_{vt}\right]$ be components (in pixels) accounting for the tangential distortion of the camera. All the intrinsic parameters of the camera are assumed to be known using any available calibration methods. Let $p_{d}^{i}={\left[\begin{array}{ccc} x_{d}^{i} & y_{d}^{i} & z_{d}^{i} \end{array}\right]}^{T}$ represent the position (in meters) of the *i*-th landmark with respect to the coordinate reference system *C* of the camera where
(4)$$p_{d}^{i}={}^{W}R_{c}\left(x_{a}^{i}-x_{c}\right)$$
and ${}^{W}R_{c}\in SO3$ is the rotation matrix, that transforms from the world coordinate reference system *W* to the coordinate reference system *C* of the camera. Recall that the rotation matrix ${}^{W}R_{c}$ is known and constant, by the assumption of using the servo-controlled camera gimbal.

### 2.3. Camera Measurement Model for the Projection of the Target

Let consider the projection of the target over the image plane of a camera. In this case, it is assumed that some visual feature points can be extracted from the target by means of some available computer vision algorithms like [35,36,37,38] or [39].

Using the pinhole model the following expression can be defined:(5)$$\begin{array}{c} h_{c}^{t}=[\begin{array}{c} u_{c}^{t}\\ v_{c}^{t} \end{array}]=\frac{1}{z_{d}^{t}}[\begin{array}{cc} \frac{f_{c}}{d_{u}} & 0\\ 0 & \frac{f_{c}}{d_{v}} \end{array}][\begin{array}{c} x_{d}^{t}\\ y_{d}^{t} \end{array}]+[\begin{array}{c} c_{u}+d_{ur}+d_{ut}\\ c_{v}+d_{vr}+d_{vt} \end{array}] \end{array}$$

Let $p_{d}^{t}={\left[\begin{array}{ccc} x_{d}^{t} & y_{d}^{t} & z_{d}^{t} \end{array}\right]}^{T}$ represent the position (in meters) of the target with respect to the coordinate reference system *C* of the camera, and:(6)$$p_{d}^{t}={}^{W}R_{c}\left(x_{t}-x_{c}\right)$$

### 2.4. Altimeter Measurement Model

Let consider an altimeter carried by the UAV. Based on altimeter readings, measurements of UAV altitude are obtained, therefore this model is simply defined by:(7)$$h_{a}=z_{c}$$

It is important to note that the only strict requirement for the proposed method is the availability of altitude measurements respect to the reference system *W*. In this case, the typical barometer-based altimeters which are equipped in most UAVs can be configured to provide such kind of measurement [40].

### 2.5. Range Measurement Model

Let consider the availability of a range sensor. Its measurements of the relative distance of the UAV with respect to the target are obtained. In this case, the measurement model is defined by:(8)$$h_{r}=\sqrt{{\left(x_{t}-x_{c}\right)}^{2}+{\left(y_{t}-y_{c}\right)}^{2}+{\left(z_{t}-z_{c}\right)}^{2}}$$

For practical implementation, several techniques like [41] or [42] can be used to obtain these kinds of measurements. On the other hand, a practical limitation for using these techniques is the requirement of a target equipped with such a device. Thus, the application of the proposed method with non-cooperative targets becomes more challenging.

## 3. Observability Analysis

In this section, the nonlinear observability properties of the proposed system are studied. Observability is an inherent property of a dynamic system and has an important role in the accuracy and stability of its estimation process. Moreover, this fact has important consequences in the convergence of the EKF-based SLAM.

In particular, it will be shown that the inclusion of the altimeter measurements improves the observability properties of the SLAM system.

A system is defined as observable if the initial state $x_{0}$, at any initial time $t_{0}$, can be determined given the state transition model $\dot{x}=f\left(x\right)$, the observation model y=h(x), and observations $z[t_{0},t]$ from time $t_{0}$ to a finite time *t*. Given the observability matrix O, a non-linear system is *locally weakly observable* if the condition rank(O)=dim(x) is verified [43].

### 3.1. Observability Matrix

An observability matrix O can be constructed in the following manner:(9)$$O={\left[\begin{array}{ccccccccc} \frac{\partial (L_{f}^{0}\left(h_{c}^{i}\right))}{\partial x} & \frac{\partial (L_{f}^{1}\left(h_{c}^{i}\right))}{\partial x} & \cdots & \frac{\partial (L_{f}^{0}\left(h_{c}^{t}\right))}{\partial x} & \frac{\partial (L_{f}^{1}\left(h_{c}^{t}\right))}{\partial x} & \frac{\partial (L_{f}^{0}\left(h_{a}\right))}{\partial x} & \frac{\partial (L_{f}^{1}\left(h_{a}\right))}{\partial x} & \frac{\partial (L_{f}^{0}\left(h_{r}\right))}{\partial x} & \frac{\partial (L_{f}^{1}\left(h_{r}\right))}{\partial x} \end{array}\right]}^{T}$$
where $L_{f}^{s}h$ represent the *s-th*-order Lie derivative [44], of the scalar field h respect to the vector field f. In this work, the rank calculation of Equation (Equation 9) was carried out numerically using MATLAB. The degree of Lie derivatives, used for computing O, was determined by gradually augmenting the matrix O with higher-order derivatives until its rank remained constant. Based on this approach, only Lie derivatives of zero and first order were needed to construct the observability matrix for all the cases.

The description of the zero and first order Lie derivatives used for constructing Equation (Equation 9) are presented in Appendix A. Using these derivatives the observability matrix n Equation (Equation 9) can be expanded as follows: (10)$$O=[\begin{array}{ccccccccc} \; & \; & \; & \; & \;\\ \;\;\;\;\;\;\;\;\;\;\;\;\;\;\;\;\;0_{2\times 6} & \; & | & -H_{c}^{i}\;{}^{W}R_{c} & 0_{2\times 3} & | & 0_{2\times 3(i-1)} & H_{c}^{i}\;{}^{W}R_{c} & 0_{2\times 3(n_{1}-i)}\\ \; & \; & \; & \; & \;\\ \;\;\;\;\;\;\;\;\;\;\;\;\;\;\;\;\;0_{2\times 6} & \; & | & H_{\mathrm{dc}}^{i} & -H_{c}^{i}\;{}^{W}R_{c} & | & 0_{2\times 3(i-1)} & -H_{\mathrm{dc}}^{i} & 0_{2\times 3(n_{1}-i)}\\ \; & \; & \; & \; & \;\\ \;\;\;\;\;\;\;\;\;\;\;\;\;\;\;\;\;\;\;\;\vdots & \; & \;\;\;\;\;\;\;\;\;\;\;\;\;\;\;\;\;\;\;\;\vdots & \; & \; & \vdots \\ \; & \; & \; & \; & \;\\ H_{c}^{t}\;{}^{W}R_{c} & 0_{2\times 3} & | & -H_{c}^{t}\;{}^{W}R_{c} & 0_{2\times 3} & | & \; & 0_{2\times 3n_{1}}\\ \; & \; & \; & \; & \;\\ -H_{\mathrm{dc}}^{t} & H_{c}^{t}\;{}^{W}R_{c} & | & H_{\mathrm{dc}}^{t} & -H_{c}^{t}\;{}^{W}R_{c} & | & \; & 0_{2\times 3n_{1}}\\ \; & \; & \; & \; & \;\\ \;\;\;\;\;\;\;\;\;\;\;\;\;\;\;\;\;0_{1\times 6} & \; & | & \left[\begin{array}{cc} 0_{1\times 2} & 1 \end{array}\right] & 0_{1\times 3} & | & \; & 0_{1\times 3n_{1}}\\ \; & \; & \; & \; & \;\\ \;\;\;\;\;\;\;\;\;\;\;\;\;\;\;\;\;0_{1\times 6} & \; & | & 0_{1\times 3} & \left[\begin{array}{cc} 0_{1\times 2} & 1 \end{array}\right] & | & \; & 0_{1\times 3n_{1}}\\ \; & \; & \; & \; & \;\\ H_{r} & 0_{1\times 3} & | & -H_{r} & 0_{1\times 3} & | & \; & 0_{1\times 3n_{1}}\\ \; & \; & \; & \; & \;\\ H_{\mathrm{dr}} & H_{r} & | & -H_{\mathrm{dr}} & -H_{r} & | & \; & 0_{1\times 3n_{1}}\\ \; & \; & \; & \; & \; \end{array}]$$

In Equations (Equation 9) and (Equation 10), Lie derivatives that belong to each kind of measurement are distributed as: first two rows (or first two elements in Equation (Equation 9)) are for monocular measurements of the landmarks; second two rows (or second two elements) are for monocular measurements of the target; third two rows (or third two elements) are for altitude measurements; and last two rows (or last two elements) are for range (UAV–target) measurements.

### 3.2. Theoretical Results

Two different cases of system configurations were analyzed. The idea is to study how the observability of the system is affected due to the availability (or unavailability) of the altimeter measurements.

#### 3.2.1. without Altimeter Measurements

In this case, considering only the respective derivatives on the observability matrix in Equation (Equation 10), the maximum rank of the observability matrix O is $rank\left(O\right)=\left(3n_{1}+12\right)-4$. In this case, $n_{1}$ is the number of measured landmarks, 12 is the number of states of the UAV–camera system and the target, and 3 is the number of states per landmark. Therefore, O will be rank deficient (rank(O)<dim(x)). The unobservable modes are spanned by the right nullspace basis $N_{1}$ of the observability matrix O.

It is straightforward to verify that the right nullspace basis of O spans for $N_{1}$, (i.e., $ON_{1}=0$). From Equation (Equation 11) it can be seen that the unobservable modes cross through all states, and thus all states are unobservable. It should be noted that adding Lie derivatives of higher-order to the observability matrix the previous result does not improve.
(11)$$N_{1}=\mathrm{null}\left(O\right)=(\frac{1}{z_{a}^{\left(i-1\right)}-z_{a}^{i}})[\begin{array}{cccc} \; & \;\\ \;\;\;x_{c}-x_{a}^{i} & \left[\begin{array}{c} \left(z_{a}^{\left(i-1\right)}-z_{a}^{i}\right)\\ \;\;\;\;\;\;\;\;\;\;0\\ \;\;\;\;\;\;\;\;\;\;0 \end{array}\right] & \left[\begin{array}{c} \;\;\;\;\;\;\;\;\;\;0\\ \left(z_{a}^{\left(i-1\right)}-z_{a}^{i}\right)\\ \;\;\;\;\;\;\;\;\;\;0 \end{array}\right] & -[\begin{array}{c} \;\;x_{c}-x_{a}^{i}\\ \;\;y_{c}-y_{a}^{i}\\ z_{c}-z_{a}^{\left(i-1\right)} \end{array}]\\ \; & \;\\ \;\;\;\;\;\;\;v_{c} & 0_{3\times 1} & 0_{3\times 1} & -v_{c}\\ \; & \;\\ ----- & -------- & -------- & ---------\\ \; & \;\\ \;\;\;\;x_{c}-x_{a}^{i} & \left[\begin{array}{c} \left(z_{a}^{\left(i-1\right)}-z_{a}^{i}\right)\\ \;\;\;\;\;\;\;\;\;\;0\\ \;\;\;\;\;\;\;\;\;\;0 \end{array}\right] & \left[\begin{array}{c} \;\;\;\;\;\;\;\;\;\;0\\ \left(z_{a}^{\left(i-1\right)}-z_{a}^{i}\right)\\ \;\;\;\;\;\;\;\;\;\;0 \end{array}\right] & -[\begin{array}{c} \;\;x_{c}-x_{a}^{i}\\ \;\;y_{c}-y_{a}^{i}\\ z_{c}-z_{a}^{\left(i-1\right)} \end{array}]\\ \; & \;\\ \;\;\;\;\;\;\;v_{c} & 0_{3\times 1} & 0_{3\times 1} & -v_{c}\\ \; & \;\\ ----- & -------- & -------- & ---------\\ \; & \;\\ \;\;\;x_{a}^{1}-x_{a}^{i} & \vdots & \vdots & -[\begin{array}{c} \;\;x_{a}^{1}-x_{a}^{i}\\ \;\;y_{a}^{1}-y_{a}^{i}\\ z_{a}^{1}-z_{a}^{\left(i-1\right)} \end{array}]\\ \;\;\;\;\;\;\;\;\;\vdots & \left[\begin{array}{c} \left(z_{a}^{\left(i-1\right)}-z_{a}^{i}\right)\\ \;\;\;\;\;\;\;\;\;\;0\\ \;\;\;\;\;\;\;\;\;\;0 \end{array}\right] & \left[\begin{array}{c} \;\;\;\;\;\;\;\;\;\;0\\ \left(z_{a}^{\left(i-1\right)}-z_{a}^{i}\right)\\ \;\;\;\;\;\;\;\;\;\;0 \end{array}\right] & \vdots \\ x_{a}^{\left(i-2\right)}-x_{a}^{i} & \vdots & \vdots & -[\begin{array}{c} \;\;x_{a}^{\left(i-2\right)}-x_{a}^{i}\\ \;\;y_{a}^{\left(i-2\right)}-y_{a}^{i}\\ z_{a}^{\left(i-2\right)}-z_{a}^{\left(i-1\right)} \end{array}]\\ \; & \;\\ x_{a}^{\left(i-1\right)}-x_{a}^{i} & \left[\begin{array}{c} \left(z_{a}^{\left(i-1\right)}-z_{a}^{i}\right)\\ \;\;\;\;\;\;\;\;\;\;0\\ \;\;\;\;\;\;\;\;\;\;0 \end{array}\right] & \left[\begin{array}{c} \;\;\;\;\;\;\;\;\;\;0\\ \left(z_{a}^{\left(i-1\right)}-z_{a}^{i}\right)\\ \;\;\;\;\;\;\;\;\;\;0 \end{array}\right] & -[\begin{array}{c} x_{a}^{\left(i-1\right)}-x_{a}^{i}\\ y_{a}^{\left(i-1\right)}-y_{a}^{i}\\ \;\;\;\;\;\;\;\;0 \end{array}]\\ \; & \;\\ \;\;\;\;\;\;\;0_{3\times 1} & \vdots & \vdots & \;\;\;\;[\begin{array}{c} \;\;\;\;\;\;\;\;\;\;0\\ \;\;\;\;\;\;\;\;\;\;0\\ \left(z_{a}^{\left(i-1\right)}-z_{a}^{i}\right) \end{array}]\\ \; & \; \end{array}]$$

#### 3.2.2. with Altimeter Measurements

When altimeter measurements are taking into account, the observability matrix in Equation (Equation 10) is rank deficient (rank(O)<dim(x)), with $rank\left(O\right)=\left(3n_{1}+12\right)-2$. In such a case, the following right nullspace basis $N_{2}$ spans the unobservable modes of O: (12)$$N_{2}=\mathrm{null}\left(O\right)={\left[\begin{array}{ccccccccc} \; & \; & \; & \; & \;\\ {\left[\begin{array}{ccc} 1 & 0 & 0 \end{array}\right]}^{T} & 0_{3\times 1} & | & {\left[\begin{array}{ccc} 1 & 0 & 0 \end{array}\right]}^{T} & 0_{3\times 1} & | & {\left[\begin{array}{ccc} 1 & 0 & 0 \end{array}\right]}^{T} & \cdots & {\left[\begin{array}{ccc} 1 & 0 & 0 \end{array}\right]}^{T}\\ \; & \; & \; & \; & \;\\ {\left[\begin{array}{ccc} 0 & 1 & 0 \end{array}\right]}^{T} & 0_{3\times 1} & | & {\left[\begin{array}{ccc} 0 & 1 & 0 \end{array}\right]}^{T} & 0_{3\times 1} & | & {\left[\begin{array}{ccc} 0 & 1 & 0 \end{array}\right]}^{T} & \cdots & {\left[\begin{array}{ccc} 0 & 1 & 0 \end{array}\right]}^{T}\\ \; & \; & \; & \; & \; \end{array}\right]}^{T}$$

It can be verified that the right nullspace basis of O spans for $N_{2}$, (i.e., $ON_{2}=0$). From Equation (Equation 12) it can be observed that the unobservable modes are related to the global position in *x* and *y* of the UAV–camera system, the landmarks, and the target. In this case, the rest of the states are observable. It should be noted that adding Lie derivatives of higher-order to the observability matrix the previous result does not improve.

#### 3.2.3. Remarks on the Theoretical Results

To interpret the former theoretical results, it is important to recall that any world-centric SLAM system is partially observable in the absence of global measurements (e.g., GPS measurements).

In this case, the SLAM system computes the position and velocity within its map, and not respect to the global reference system. Fortunately, this is not a problem for some applications that require local or relative position estimates, for instance the problem addressed in this work that implies to following a moving target.

On the other hand, it is worth noting how the simple inclusion of an altimeter improves the observability properties of the system when GPS measurements are not considered (see Table 1). It is very interesting to observe that, besides the states along the *z*-axis $\left[z_{t},z_{c},z_{a}^{i}\right]$ (as one could expect), the velocity of the camera-robot (which is global-referenced) becomes observable when altitude measurements are included. In this case, since the target is estimated respect to the camera, the global velocity of the target becomes observable.

Accordingly, it is also important to note that, since the range and monocular measurements to the target are used only for estimating the position of the target with respect to the camera-robot, these measurements affect neither the observability of the camera-robot state nor the observability of the landmarks states.

In other words, the target measurements create only a “link” to the camera-robot state that allows estimating the relative position of the target but does not provide any information about the state of the camera-robot, and for this reason, they are not included in the observability analysis.

Later, it will be seen how the target position is used for improving the initialization of nearby landmarks, which in turn improves the robustness and accuracy of the system.

A final but very important remark is to consider that the order of Lie derivatives and the rank calculation of Equation (Equation 9) were determined numerically, but not analytically. Therefore, there is still a chance, in rigorous terms, that a subset of the unobservable states determined by the analysis is in reality observable (see [43]).

## 4. Ekf-Based Slam

In this work, the standard EKF-based SLAM scheme [45,46] is used to estimate the system state in Equation (Equation 2). The architecture of the proposed system is shown in Figure 2.

From Equation (Equation 1), the following discrete system can be defined:(13)$$x_{k}=f\left(x_{k-1},n_{k-1}\right)=[\begin{array}{c} \;x_{t}k\\ \;v_{t}k\\ \;x_{c}k\\ \;v_{c}k\\ {}^{e}x_{a}^{j}k\\ {}^{p}x_{a}^{n}k \end{array}]=[\begin{array}{c} x_{t}k-1+\left(v_{t}k-1\right)\Delta t\\ \;\;\;v_{t}k-1+\eta_{t}k-1\\ x_{c}k-1+\left(v_{c}k-1\right)\Delta t\\ \;\;\;v_{c}k-1+\eta_{c}k-1\\ \;\;\;\;\;\;\;\;{\;}^{e}x_{a}{k-1}^{j}\\ \;\;\;\;\;\;\;\;{\;}^{p}x_{a}{k-1}^{n} \end{array}]$$
(14)$$n_{k}=[\begin{array}{c} \eta_{t}k\\ \eta_{c}k \end{array}]=[\begin{array}{c} a_{t}\Delta t\\ a_{c}\Delta t \end{array}]$$

From Equations (Equation 3), (Equation 5), (Equation 7) and (Equation 8), the system measurements are defined as follows: (15)$$z_{k}=h\left(x_{k},r_{k}\right)=[\begin{array}{c} {\;}^{e}h_{c}^{j}k+{}^{e}r_{c}k^{j}\\ {}^{p}h_{c}^{n}k+{}^{p}r_{c}k^{n}\\ \;\;\;h_{t}k+r_{t}k\\ \;\;\;{h_{a}}_{k}+{r_{a}}_{k}\\ \;\;\;{h_{r}}_{k}+{r_{r}}_{k} \end{array}]$$
(16)$$r_{k}=[\begin{array}{c} {}^{e}r_{c}k^{j}\\ {}^{p}r_{c}k^{n}\\ \;r_{t}k\\ \;{r_{a}}_{k}\\ \;{r_{r}}_{k} \end{array}]$$

Let ${}^{e}x_{a}^{j}={\left[\begin{array}{ccc} {}^{e}x_{a}^{j} & {}^{e}y_{a}^{j} & {}^{e}z_{a}^{j} \end{array}\right]}^{T}$ be the *j-th* landmark defined by its Euclidean parametrization. Let ${}^{p}x_{a}^{n}={\left[\begin{array}{cccccc} {{}^{p}x_{c}^{n}}_{o} & {{}^{p}y_{c}^{n}}_{o} & {z_{c}^{n}}_{o} & {}^{p}\theta_{a}^{n} & {}^{p}\phi_{a}^{n} & {}^{p}\rho_{a}^{n} \end{array}\right]}^{T}$ be the *n-th* landmark defined by its inverse of the depth parametrization, $j=1,\ldots ,n_{2}$, where $n_{2}$ is the number of landmarks with Euclidean parametrization, $n=1,\ldots ,n_{3}$, where $n_{3}$ is the number of landmarks with inverse of the depth parametrization, and $n_{1}=n_{2}+n_{3}$.

Let ${{}^{p}x_{c}}_{o}^{n}={\left[\begin{array}{ccc} {{}^{p}x_{c}^{n}}_{o} & {{}^{p}y_{c}^{n}}_{o} & {{}^{p}z_{c}^{n}}_{o} \end{array}\right]}^{T}$ represent the position (in meters) of the camera when the feature ${}^{p}x_{a}^{n}$ was observed for the first time. Let ${}^{p}\theta_{a}^{n}$ and ${}^{p}\phi_{a}^{n}$ be azimuth and elevation respectively (respect to the global reference frame W). Let ${}^{p}\rho_{a}^{n}=\frac{1}{{}^{p}d^{n}}$ be the inverse of the depth ${}^{p}d^{n}$. Let ${}^{e}h_{c}^{j}$ be the projection over the image plane of a camera of the *j-th* landmark. Let ${}^{p}h_{c}^{n}$ be the projection over the image plane of a camera of the *n-th* landmark.

In Equation (14), $a_{t}$ and $a_{c}$ are zero-mean Gaussian noise representing unknown linear accelerations dynamics. Moreover, $n_{k}∼N\left(0,Q_{k}\right)$, $r_{k}∼N\left(0,R_{k}\right)$ are uncorrelated noise vectors affecting respectively the system dynamics and the system measurements. Let *k* be the sample step, and Δt is the time differential. It is important to note that the proposed scheme does not depend on a specific aircraft dynamical model.

The EKF prediction equations are:(17)$${\hat{x}}_{k}^{-}=f\left({\hat{x}}_{k-1},0\right)$$
(18)$$P_{k}^{-}=A_{k}P_{k-1}A_{k}^{T}+W_{k}Q_{k-1}W_{k}^{T}$$

The EKF update equations are:(19)$${\hat{x}}_{k}={\hat{x}}_{k}^{-}+K_{k}\left(z_{k}-h\left({\hat{x}}_{k}^{-},0\right)\right)$$
(20)$$P_{k}=\left(I-K_{k}C_{k}\right)P_{k}^{-}$$
with (21)$$K_{k}=P_{k}^{-}C_{k}^{T}{\left(C_{k}P_{k}^{-}C_{k}^{T}+V_{k}R_{k}V_{k}^{T}\right)}^{-1}$$
and (22)$$\begin{array}{c} A_{k}=\frac{\partial f}{\partial x}\left({\hat{x}}_{k-1},0\right)\;C_{k}=\frac{\partial h}{\partial x}\left({\hat{x}}_{k}^{-},0\right)\\ W_{k}=\frac{\partial f}{\partial n}\left({\hat{x}}_{k-1},0\right)\;V_{k}=\frac{\partial h}{\partial r}\left({\hat{x}}_{k}^{-},0\right) \end{array}$$

Let K be the Kalman gain, and let P be the system covariance matrix.

### 4.1. Map Features Initialization

The system state x is augmented with new map features when a landmark is observed for the first time. The landmark can be initialized in one of two different parametrizations: (i) Euclidean parametrization and (ii) Inverse depth parametrization, depending on how close this landmark is from the target. Since the target is assumed to move over the ground, the general idea is to use the range information provided by the target to infer the initial depth of the landmarks near to the target. In this case, it will be assumed that the landmarks near the target lie at a similar altitude, situation encountered in most of the applications. It is important to recall that the initialization of landmarks plays a fundamental role in the robustness and convergence of the EKF-based SLAM.

#### 4.1.1. Initialization of Landmarks near to the Target

A landmark is initialized with a Euclidean parameterization if it is supposed arbitrarily near the target. This assumption is made if the landmark is within a selected area of the image (see Section 4.1.4). In this case, the landmark can be initialized with the information given by the range measurement between the UAV and the target, which is assumed to be approximately equal to the depth that the landmark has respect to the camera.

Therefore, the following equation is defined:(23)$${}^{e}x_{a}^{j}={{\hat{x}}_{c}}_{k}+h_{r}[\begin{array}{c} \cos({}^{e}\theta_{a}^{j})\cos({}^{e}\phi_{a}^{j})\\ \sin({}^{e}\theta_{a}^{j})\cos({}^{e}\phi_{a}^{j})\\ \;\;\;\;\;\;\;\;\sin({}^{e}\phi_{a}^{j}) \end{array}]$$
where ${{\hat{x}}_{c}}_{k}$ is the estimated position of the camera when the feature ${}^{e}x_{a}^{j}$ was observed for first time, and
(24)$$\left[\begin{array}{c} {}^{e}\theta_{a}^{j}\\ {}^{e}\phi_{a}^{j} \end{array}]=[\begin{array}{c} \arctan2({{}^{e}g_{a}}_{y}^{j},{{}^{e}g_{a}}_{x}^{j})\\ \arctan2({{}^{e}g_{a}}_{z}^{j},\sqrt{{\left({{}^{e}g_{a}}_{x}^{j}\right)}^{2}+{\left({{}^{e}g_{a}}_{y}^{j}\right)}^{2}}) \end{array}\right]$$
with ${{}^{e}g_{a}}^{j}={\left[\begin{array}{ccc} {{}^{e}g_{a}}_{x}^{j} & {{}^{e}g_{a}}_{y}^{j} & {{}^{e}g_{a}}_{z}^{j} \end{array}\right]}^{T}{=}^{W}R_{c}^{T}{\left[\begin{array}{ccc} {}^{e}u_{c}^{j} & {}^{e}v_{c}^{j} & f_{c} \end{array}\right]}^{T}$. Where, ${[}^{e}u_{c}^{j}$,${}^{e}v_{c}^{j}]$ define the coordinates (in pixels) of the projection of the *j*-th landmark over the image of the camera. In case of a landmark with Euclidean parameterization, the projection over the image plane of a camera is defined:(25)$$\begin{array}{c} {}^{e}h_{c}^{j}=[\begin{array}{c} {}^{e}u_{c}^{j}\\ {}^{e}v_{c}^{j} \end{array}]=\frac{1}{{}^{e}z_{d}^{j}}[\begin{array}{cc} \frac{f_{c}}{d_{u}} & 0\\ 0 & \frac{f_{c}}{d_{v}} \end{array}][\begin{array}{c} {}^{e}x_{d}^{j}\\ {}^{e}y_{d}^{j} \end{array}]+[\begin{array}{c} c_{u}+d_{ur}+d_{ut}\\ c_{v}+d_{vr}+d_{vt} \end{array}] \end{array}$$
with (26)$${}^{e}p_{d}^{j}=[\begin{array}{c} {}^{e}x_{d}^{j}\\ {}^{e}y_{d}^{j}\\ {}^{e}z_{d}^{j} \end{array}]={}^{W}R_{c}({}^{e}x_{a}^{j}-x_{c})$$

#### 4.1.2. Initialization of Landmarks Far from the Target

A landmark is initialized with an inverse depth parametrization if it is supposed arbitrarily far from the target. This assumption is made if the landmark is outside a selected area of the image (see Section 4.1.4). In this case, ${{}^{p}x_{c}}_{o}^{n}$ is given for the estimated position of the camera ${\hat{x}}_{c}$${}_{k}$ when the feature ${}^{p}x_{a}^{n}$ was observed for the first time. Furthermore, the following equation is defined:(27)$$\left[\begin{array}{c} {}^{p}\theta_{a}^{n}\\ {}^{p}\phi_{a}^{n} \end{array}]=[\begin{array}{c} \arctan2({{}^{p}g_{a}}_{y}^{n},{{}^{p}g_{a}}_{x}^{n})\\ \arctan2({{}^{p}g_{a}}_{z}^{n},\sqrt{{\left({{}^{p}g_{a}}_{x}^{n}\right)}^{2}+{\left({{}^{p}g_{a}}_{y}^{n}\right)}^{2}}) \end{array}\right]$$
with ${{}^{p}g_{a}}^{n}={\left[\begin{array}{ccc} {{}^{p}g_{a}}_{x}^{n} & {{}^{p}g_{a}}_{y}^{n} & {{}^{p}g_{a}}_{z}^{n} \end{array}\right]}^{T}{=}^{W}R_{c}^{T}{\left[\begin{array}{ccc} {}^{p}u_{c}^{n} & {}^{p}v_{c}^{n} & f_{c} \end{array}\right]}^{T}$. Where, ${[}^{p}u_{c}^{n}$,${}^{p}v_{c}^{n}]$ define the coordinates (in pixels) of the projection of the *n*-th landmark over the image of the camera. ${}^{p}\rho_{a}^{n}$ is initialized as it is shown in [47]. In case of a landmark with inverse depth parametrization, the projection over the image plane of a camera is defined by:(28)$$\begin{array}{c} {}^{p}h_{c}^{n}=[\begin{array}{c} {}^{p}u_{c}^{n}\\ {}^{p}v_{c}^{n} \end{array}]=\frac{1}{{}^{p}z_{d}^{n}}[\begin{array}{cc} \frac{f_{c}}{d_{u}} & 0\\ 0 & \frac{f_{c}}{d_{v}} \end{array}][\begin{array}{c} {}^{p}x_{d}^{n}\\ {}^{p}y_{d}^{n} \end{array}]+[\begin{array}{c} c_{u}+d_{ur}+d_{ut}\\ c_{v}+d_{vr}+d_{vt} \end{array}] \end{array}$$
with (29)$${}^{p}p_{d}^{n}=[\begin{array}{c} {}^{p}x_{d}^{n}\\ {}^{p}y_{d}^{n}\\ {}^{p}z_{d}^{n} \end{array}]={}^{W}R_{c}({{}^{p}x_{c}}_{o}^{n}+\frac{1}{{}^{p}\rho_{a}^{n}}[\begin{array}{c} \cos({}^{p}\theta_{a}^{n})\cos({}^{p}\phi_{a}^{n})\\ \sin({}^{p}\theta_{a}^{n})\cos({}^{p}\phi_{a}^{n})\\ \;\;\;\;\;\;\;\;\sin({}^{p}\phi_{a}^{n}) \end{array}]-x_{c})$$

#### 4.1.3. State Augmentation

To initialize a new landmark, the system state x must be augmented by $x={\left[\begin{array}{ccccccc} x_{t} & v_{t} & x_{c} & v_{c} & {}^{e}x_{a}^{j} & {}^{p}x_{a}^{n} & x_{a}^{new} \end{array}\right]}^{T}$, being $x_{a}^{new}$ the new landmark which is initialized by either the Euclidean or the inverse depth parametrization. Thus, a new covariance matrix $P_{new}$ is computed by: (30)$$P_{new}=\Delta J[\begin{array}{cc} P & 0\\ 0 & R^{i} \end{array}]\Delta J^{T}$$
where $R^{i}$ is the measurement noise covariance matrix, ΔJ is the Jacobian $\frac{\partial h(x)}{\partial x}$, and h(x) is the landmark initialization function.

#### 4.1.4. Landmarks Selection Method

To determine whether a landmark is initialized with Euclidean or inverse depth parametrization, it should be determined arbitrarily if the landmark is considered near or far from the target. To achieve this objective the following heuristic is used (see Figure 3): (1) firstly, a spherical area centered on the target of radius $r_{w}$ is defined; (2) then, the radius $r_{c}$ of the projected spherical area in the camera is estimated; and (3) the landmarks whose projection in the camera are within the projected spherical area ${(}^{i}d_{t}\leq r_{c})$ are considered near to the target and thus, they are initialized with Euclidean parameterization (see Section 4.1.1). Otherwise ${(}^{i}d_{t}>r_{c})$, landmarks are considered far from the target, and they are initialized with inverse depth parametrization (see Section 4.1.2) where ${}^{i}d_{t}=\sqrt{{\left(u_{c}^{t}-u_{c}^{i}\right)}^{2}+{\left(v_{c}^{t}-v_{c}^{i}\right)}^{2}}$.

Here, $r_{c}$ is estimated as follows:(31)$$r_{c}=\sqrt{{\left(u_{c}^{t}-u_{c}^{r}\right)}^{2}+{\left(v_{c}^{t}-v_{c}^{r}\right)}^{2}}$$
where (32)$$\begin{array}{c} \left[\begin{array}{c} u_{c}^{r}\\ v_{c}^{r} \end{array}]=\frac{1}{z_{d}^{r}}[\begin{array}{cc} \frac{f_{c}}{d_{u}} & 0\\ 0 & \frac{f_{c}}{d_{v}} \end{array}][\begin{array}{c} x_{d}^{r}\\ y_{d}^{r} \end{array}]+[\begin{array}{c} c_{u}+d_{ur}+d_{ut}\\ c_{v}+d_{vr}+d_{vt} \end{array}\right] \end{array}$$
with (33)$$p_{d}^{r}=[\begin{array}{c} x_{d}^{r}\\ y_{d}^{r}\\ z_{d}^{r} \end{array}]={}^{W}R_{c}({\hat{x}}_{t}+\eta -{\hat{x}}_{c})$$
and $\eta ={\left[\begin{array}{ccc} r_{w} & 0 & 0 \end{array}\right]}^{T}$.

## 5. Target Tracking Control

To allow a UAV to follow a target a high-level control scheme is presented. Firstly, the kinematic model of the UAV is defined:(34)$$\begin{array}{c} {\dot{x}}_{q}=v_{x}\cos\left(\psi_{q}\right)-v_{y}\sin\left(\psi_{q}\right)\\ {\dot{y}}_{q}=v_{x}\sin\left(\psi_{q}\right)+v_{y}\cos\left(\psi_{q}\right)\\ {\dot{z}}_{q}=v_{z}\\ {\dot{\psi }}_{q}=\omega \end{array}$$

Let $x_{q}={\left[\begin{array}{ccc} x_{q} & y_{q} & z_{q} \end{array}\right]}^{T}$ represent the UAV position respect to the reference system *W* (in m). Let $\left(v_{x},v_{y}\right)$ represent the UAV linear velocity along the *x* and *y* axis (in m/s) respect to the reference system *Q*. Let $v_{z}$ represent the UAV linear velocity along the *z* (vertical) axis (in m/s) respect to the reference system *W*. Let ψ represent the UAV yaw angle respect to *W* (in radians); and let ω (in radians/s) is the first derivative of ψ (angular velocity).

The proposed high-level control scheme is intended to maintain a stable flight formation of the UAV with respect to the target, by generating velocity commands to the UAV. In this case, it is assumed that a low-level (i.e., actuator-level) velocity control scheme exists, like [48] or [49], that drives the velocities $\left[v_{x},v_{y},v_{z},\omega \right]$ commanded by a high-level control.

### Visual Servoing and Altitude Control

By deriving Equation (Equation 5) the following expression can be obtained, neglecting the dynamics of the tangential and radial distortion components, taking into account that $x_{q}=x_{c}{-}^{q}d_{c}$, where ${}^{q}d_{c}$ is the translation vector (in meters) from the coordinate reference system *Q* to the coordinate reference system *C*, and assuming ${}^{q}d_{c}$ to be known and constant:(35)$$[\begin{array}{c} {\dot{u}}_{c}^{t}\\ {\dot{v}}_{c}^{t} \end{array}]=J_{c}^{t}{}^{W}R_{c}\left({\dot{x}}_{t}-{\dot{x}}_{q}\right)$$
with (36)$$J_{c}^{t}=[\begin{array}{ccc} \frac{f_{c}}{d_{u}z_{d}^{t}} & 0 & -\frac{\left(u_{c}^{t}-c_{u}-d_{ur}-d_{ut}\right)}{z_{d}^{t}}\\ 0 & \frac{f_{c}}{d_{v}z_{d}^{t}} & -\frac{\left(v_{c}^{t}-c_{v}-d_{vr}-d_{vt}\right)}{z_{d}^{t}} \end{array}]$$

Furthermore, an altitude differential $\lambda_{z}$ to be maintained from the UAV to the target is defined:(37)$$\lambda_{z}=z_{q}-z_{t}$$

Now, differentiating Equation (Equation 37):(38)$${\dot{\lambda }}_{z}={\dot{z}}_{q}-{\dot{z}}_{t}$$

Taking Equations (Equation 34), (Equation 35) and (Equation 38), the following dynamics is defined:(39)$$\dot{\lambda }=g+Bu$$
where (40)$$\begin{array}{c} \lambda =[\begin{array}{c} u_{c}^{t}\\ v_{c}^{t}\\ \lambda_{z}\\ \psi_{q} \end{array}]\;g=[\begin{array}{cc} J_{c}^{t}{}^{W}R_{c} & 0_{2\times 1}\\ \;\;\;\;c_{1} & 0_{2\times 1} \end{array}][\begin{array}{c} {\dot{x}}_{t}\\ 0 \end{array}]\;B=-[\begin{array}{cc} J_{c}^{t}{}^{W}R_{c} & 0_{2\times 1}\\ \;\;\;\;c_{1} & c_{2} \end{array}]\Omega \;u=[\begin{array}{c} v_{x}\\ v_{y}\\ v_{z}\\ \omega \end{array}] \end{array}$$
with (41)$$\begin{array}{c} c_{1}=[\begin{array}{ccc} 0 & 0 & -1\\ 0 & 0 & 0 \end{array}]\;c_{2}=[\begin{array}{c} \;0\\ -1 \end{array}]\;\Omega =[\begin{array}{cccc} \cos(\psi_{q}) & -\sin(\psi_{q}) & 0 & 0\\ \sin(\psi_{q}) & \cos(\psi_{q}) & 0 & 0\\ \;\;\;\;\;0 & 0 & 1 & 0\\ \;\;\;\;\;0 & 0 & 0 & 1 \end{array}] \end{array}$$

It will be assumed that disturbances, as well as unmodeled uncertainty, enters the system through the input. In this case, Equation (Equation 39) is redefined as:(42)$$\dot{\lambda }=g+\Delta_{g}+Bu$$
where the term $\Delta_{g}$ (representing the unknown disturbances and uncertainties) satisfies $\|\Delta_{g}\|\leq \epsilon$, where ϵ is a positive constant, so it is assumed to be bounded. Based on the dynamics in Equation (Equation 39), a robust controller is designed using the sliding mode control technique [50]. For the controller, the state-feedback is obtained from the SLAM estimator presented in Section 4. In this case, it is assumed that the UAV yaw angle is obtained directly from the AHRS device. The architecture of closed-loop system is show in Figure 4.

First, the transformation ${\hat{x}}_{q}={\hat{x}}_{c}-{}^{q}d_{c}^{j}$ is defined, to obtain the UAV estimated position in terms of the reference system *Q*.

To design the controller, the following expression is defined:(43)$$s_{\lambda }=e_{\lambda }+K_{1}\int_{0}^{t}e_{\lambda }dt$$
where $K_{1}$ is a positive definite diagonal matrix, and $e_{\lambda }=\hat{\lambda }-\lambda_{d}$, and $\lambda_{d}$ is the reference signal vector. By deriving Equation (Equation 43) and substituting in Equations (Equation 39), the following expression is obtained:(44)$${\dot{s}}_{\lambda }=-{\dot{\lambda }}_{d}+K_{1}e_{\lambda }+\hat{g}+\Delta_{g}+\hat{B}u$$

For the former dynamics, the following control law is proposed:(45)$$u={\hat{B}}^{-1}({\dot{\lambda }}_{d}-K_{1}e_{\lambda }-\hat{g}-K_{2}\mathrm{sign}\left(s_{\lambda }\right))$$
where $K_{2}$ is a positive definite diagonal matrix. Appendix B shows the proof of the existence of ${\hat{B}}^{-1}$.

A Lyapunov candidate function is defined to prove the stability of the closed-loop dynamics: (46)$$V_{\lambda }=\frac{1}{2}s_{\lambda }^{T}s_{\lambda }$$
with its corresponding derivative:(47)$$\begin{array}{c} {\dot{V}}_{\lambda }=s_{\lambda }^{T}{\dot{s}}_{\lambda }=s_{\lambda }^{T}(-{\dot{\lambda }}_{d}+K_{1}e_{\lambda }+\hat{g}+\Delta_{g}+\hat{B}u) \end{array}$$

So, by substituting Equation (Equation 45) in Equation (Equation 47), the following expression can be obtained:(48)$$\begin{array}{c} {\dot{V}}_{\lambda }=s_{\lambda }^{T}(\Delta_{g}-K_{2}\mathrm{sign}\left(s_{\lambda }\right))\\ \leq \|s_{\lambda }\left\|\right\|\Delta_{g}\|-s_{\lambda }^{T}\left\|K_{2}\right\|\mathrm{sign}\left(s_{\lambda }\right)\\ \leq \|s_{\lambda }\|\epsilon -\alpha s_{\lambda }^{T}\mathrm{sign}\left(s_{\lambda }\right)\\ \leq \|s_{\lambda }\left\|\epsilon -\right\|s_{\lambda }\|\alpha \\ \leq \|s_{\lambda }\|\left(\epsilon -\alpha \right) \end{array}$$
where $\alpha =\lambda_{min}\left(K_{2}\right)$. Therefore, if α is chosen such that α>ϵ, then ${\dot{V}}_{\lambda }$ will be negative definite. In this case, the dynamics defining the flight formation will reach the surface $s_{\lambda }=0$ and will remain there in a finite time.

## 6. Experimental Results

To validate the performance of the proposed method, simulations and experiments with real data have been carried out.

### 6.1. Simulations

#### 6.1.1. Simulation Setup

The proposed cooperative UAV–Target visual-SLAM method is validated through computer simulations. For this purpose, a Matlab${}^{®}$ implementation of the proposed scheme was used. The simulated UAV–Target environment is composed of 3D landmarks, which are randomly distributed over the ground. In this case, a UAV equipped with the required sensors is simulated. To include uncertainty into the simulations, the following Gaussian noise is added to measurements: for camera measurements $\sigma_{c}=4$ pixels; for altimeter measurements $\sigma_{a}=25$ cm; and for range sensor measurements $\sigma_{r}=25$ cm. All measurements are emulated to be acquired with a frequency of 10 Hz. The magnitude of the camera noise is bigger than the typical noise of real monocular measurement. In this way, it is intended to consider, in addition to the imperfection of the sensor, the errors in camera orientation due to the gimbal assumption. In simulations, the target was moved along a predefined trajectory.

#### 6.1.2. Convergence and Stability Tests

The objective of the test presented in this subsection is to show how the robustness of the SLAM system takes advantage of the inclusion of the altimeter measurements. In other words, both observability conditions described in Section 3 (with and without altimeter measurements) are tested. For this test, a control system is assumed to exist able to maintain the target tracking by the UAV.

It is well known that the initial conditions of the landmarks play an important role in the convergence and stability of SLAM systems. Therefore, a good way to evaluate the robustness of the SLAM system is to force bad initial conditions for the landmarks states. This means that (only for this test) the proposed initialization technique, described in Section 4.1, will no be used. Instead, the initial states of the landmarks $x_{a}^{new}$, will be randomly determined using different standard deviations for the error position. Note that in this case, the initial conditions of ${\hat{x}}_{t}$, ${\hat{x}}_{c}$, ${\hat{v}}_{t}$, ${\hat{v}}_{c}$ are assumed to be exactly known.

Three different conditions of initial error are considered: $\sigma_{a}=\left\{1,2,3\right\}$ meters, with a continuous uniform distribution. Figure 5 shows the actual trajectories followed by the target and the UAV.

Figure 6 shows the results of the tests. The estimated positions of the UAV are plotted for each reference axis (row plots), and each column of plots shows the results obtained from each observability case. The results of the estimated state of the target are very similar to those presented for the UAV and, therefore, are omitted.

Table 2 summarizes the Mean Squared Error (MSE) of the estimated positions obtained for both the target and the UAV.

Taking a closer look at Figure 6 and Table 2, it can be observed that both, the observability property and the initial conditions, play a preponderant role in the convergence and stability of the EKF-SLAM. For several applications, the initial position of the UAVs can be assumed to be known. However, in SLAM, the position of the map features must be estimated online. That confirms the great importance of using good features initialization techniques in visual-SLAM; and, as it can be expected, the better observability properties the better performance of the EKF-SLAM system, indeed.

#### 6.1.3. Comparative Study

In this subsection a comparative study between the proposed monocular-based SLAM method and the following methods is presented,

(1)Monocular SLAM(2)Monocular SLAM with anchors.(3)Monocular SLAM with inertial measurements.(4)Monocular SLAM with altimeter.(5)Monocular SLAM with a cooperative target: without target-based initialization.

There are some remarks about the methods used in the comparison. The method (1) is the approach described in [47]. This method is included only as a reference to highlight that purely monocular methods cannot retrieve the metric scale of the scene. The method (2) is similar to the previous method. But in this case, to set the metric scale of the estimates, the position of a subset of the landmarks seen in the first frame is assumed to be perfectly known (anchors). The method (3) is the approach described in [12]. In this case, inertial measurements obtained from an inertial measurement unit (IMU) are fused into the system. For this IMU-based method, it is assumed that the alignment of the camera and the inertial measurement unit is perfectly known; the dynamic error bias of the accelerometers is neglected as well. The method (4) is the approach proposed in [13]. In this case, altitude measurements given by an altimeter are fused into the monocular SLAM system. The method (5) is a variation of the proposed method. In this case, the landmark initialization technique proposed in Section 4.1 is not used, and instead only the regular initialization method is used. Therefore, this variation of the proposed method is included in the comparative study to highlight the advantage of the proposed cooperative-based initialization technique.

It is worth pointing out that all the methods (included the proposed method) use the same hypothetical initial depth for the landmarks without a priori inference of their position. Also for the comparative study, a control system is assumed to exist able to maintain the target tracking by the UAV.

##### First Comparison Test

Using the simulation setup illustrated in Figure 5, the performance of all the methods were tested for estimating the position of the camera-robot and the map of landmarks.

Figure 7 shows the results obtained from each method when estimating the position of the UAV. Figure 8 shows the results obtained from each method when estimating the velocity of the UAV. For the sake of clarity, the results of Figure 7 and Figure 8 are shown in two columns of plots. Each row of plots represents a reference axis.

Table 3 summarizes the Mean Squared Error (MSE) for the estimated relative position of the UAV expressed in each one of the three axes. Table 4 summarizes the Mean Squared Error (MSE) for the estimated position of the landmarks expressed in each one of the three axes.

##### Second Comparison Test

In this comparison test, the performance of all the methods was tested for recovering the metric scale of the estimates. For this test, the target and the UAV follow a circular trajectory for 30 s. During the flight, the altitude of the UAV was changed (see Figure 9). In this case, it is assumed that all the landmarks on the map are seen from the first frame and that they are kept in the camera field of view throughout the simulation.

The scale factor *s* is given by [51]:(49)$$s=\frac{d_{real}}{d_{est}}$$
where $d_{real}$ is the real distance, and $d_{est}$ is the estimated distance. For the monocular SLAM problem, there exist different kind of distances and lots of data for real (and estimated) distances: distances between camera and landmarks, distances between landmarks, distances defined by the positions of the camera in time periods (camera trajectory), among other distances. Therefore, in practice, there is not such a standard convention for determining the metric scale. But in general, for determining the scale, the averages of multiple real and estimated distances are considered. In this work, authors propose to use the following approximation, which averages all the distances among the map features.
(50)$$\begin{array}{c} d_{real}=\frac{1}{\sum_{k=1}^{n-1}\left(n-k\right)}\sum_{i=1}^{n}\sum_{j=i+1}^{n}d^{ij}\;d_{est}=\frac{1}{\sum_{k=1}^{n-1}\left(n-k\right)}\sum_{i=1}^{n}\sum_{j=i+1}^{n}{\hat{d}}^{ij} \end{array}$$

Let $d^{ij}$ represent the actual distance of the *i*-th landmark respect to the *j*-th landmark. Let ${\hat{d}}^{ij}$ represent the estimated distance of the *i*-th landmark respect to the camera *j*-th landmark, and let *n* be the total number of landmarks. From (Equation 49), if the metric scale is perfectly recovered then s=1.

For this test, an additional method has been added for comparison purposes. The *Monocular SLAM with altimeter (Loosely-coupled)* explicitly computes the metric scale by using the ratio between the altitude obtained from an altimeter, and the unscaled altitude obtained from a purely monocular SLAM system. The computed metric scale is used then for scaling the monocular SLAM estimates.

Case 1: The UAV follows a circular flight trajectory while varying its altitude (see Figure 9, upper plot). In this case, the UAV gets back to fly over its initial position, and thus, the initial landmarks are seen again (loop-closure).

Figure 9 (lower plot) shows the evolution of the metric scale obtained for each method. In this case, for each method, the metric scale converged to a value, and remains almost constant. Even the monocular SLAM method (yellow) which does not incorporate any metric information, and the monocular SLAM with anchors (green) that only includes metric information at the beginning of the trajectory, exhibit the same behavior. It is important to note that this is the expected behavior since the camera-robot is following a circular trajectory with loop closure where the initial low-uncertainty landmarks are revisited.

Case 2: The UAV follows the same flight trajectory illustrated in Figure 5. In this case, the UAV drifts apart from its initial position, and thus, the initial landmarks are never seen again.

Figure 10 (upper plot) shows the evolution of the metric scale obtained for each method. In this case, the monocular SLAM method (yellow) was manually tuned to have a good initial metric scale. The initial conditions of the other methods are alike as those of the *Case 1*, but the vertical limits of the plot have been adjusted for better visualization. Figure 10 (middle and lower plots respectively) shows the Euclidean mean error in position for the camera-robot $e_{c}$ and the Euclidean mean error in position for the landmarks $e_{a}$ for each method, where
$$\begin{array}{cc} e_{c} & =\sqrt{{\left(x_{c}-{\hat{x}}_{c}\right)}^{2}+{\left(y_{c}-{\hat{y}}_{c}\right)}^{2}+{\left(z_{c}-{\hat{z}}_{c}\right)}^{2}}\\ e_{a} & =\sqrt{{\left(\frac{1}{n}\sum_{i=1}^{n}x_{a}^{i}-{\hat{x}}_{a}^{i}\right)}^{2}+{\left(\frac{1}{n}\sum_{i=1}^{n}y_{a}^{i}-{\hat{y}}_{a}^{i}\right)}^{2}+{\left(\frac{1}{n}\sum_{i=1}^{n}z_{a}^{i}-{\hat{z}}_{a}^{i}\right)}^{2}} \end{array}$$

Observing Figure 10, as could be expected, for the methods that continuously incorporate metric information into the system through additional sensors, the metric scale converges to a value, and remains approximately constant (after time > 100 s). On the other hand, the methods that do not continuously incorporate metric information (monocular SLAM and monocular SLAM with anchors), exhibit a drift in the metric scale. As one could also expect in SLAM without loop-closure, all the methods present some degree of drifting in position, both for the robot-camera trajectory and the landmarks. The above reasoning is independent of the drift in metric scale (the methods with low drift in scale also present drift in position). Evidently, it is convenient to maintain a low error/drift in scale, because it affects the error/drift in position.

It is interesting to note that the loosely-coupled method (purple) appears to be extremely sensitive to measurements noise. In this case, the increasing “noise” in error position is because the scale correction-ratio increases as the error in the scale of the purely monocular SLAM (yellow) also increases. In other terms, the signal-to-noise ratio (S/N) increases. Surely some adaptations can be done, as filtering the computed metric scale, but a trade-off would be introduced between the time of convergence and the reduction of noise effects.

#### 6.1.4. Estimation and Control Test

A set of simulations were also carried out to test in a closed-loop manner the proposed control scheme. In this case, the estimates obtained from the proposed visual-based SLAM estimation method are used as feedback to the control scheme described in Section 5. The value of the vector $\lambda_{d}$, that defines the desired values of the servo visual and altitude control, is: $\lambda_{d}={\left[\begin{array}{cccc} 0, & 0, & 7+\sin(t\cdot 0.05), & atan2({\hat{y}}_{q},{\hat{x}}_{q}) \end{array}\right]}^{T}$. Those values for the desired control mean that the UAV has to remain flying exactly over the target at a varying relative altitude.

Figure 11 shows the evolution of the error respect to the desired values $\lambda_{d}$. In all the cases, note that the errors are bounded after an initial transient period.

Figure 12 shows the real and estimated position of the target and the UAV.

Table 5 summarizes the Mean Squared Error (MSE), expressed in each of the three axes, for the estimated position of: (i) the target, (ii) the UAV, and (iii) the landmarks.

Table 6 summarizes the Mean Squared Error (MSE) for the initial hypotheses of landmarks depth MSEd. Furthermore, Table 6 shows the Mean Squared Error (MSE) for the estimated position of landmarks, expressed in each of the three axes. In this case, since the landmarks near to the target are initialized with a small error, its final position is better estimated. Once again, this result shows the importance of the initialization process of landmarks in SLAM.

According to the above results, it can be seen that the proposed estimation method has a good performance to estimate the position of the UAV and the target. It can also be seen that the control system was able to maintain a stable flight formation along with all the trajectory respect to the target, using the proposed visual-based SLAM estimation system as a feedback.

### 6.2. Experiments with Real Data

To test the proposed cooperative UAV–Target visual-SLAM method, an experiment with real data was carried out. In this case, a Parrot Bebop 2${}^{®}$ quadcopter [33] (see Figure 13) was used for capturing real data with its sensory system.

The set of sensors of the Bebop 2 that were used in experiments consists of (i) a camera with a wide-angle lens and (ii) a barometer-based altimeter. The drone camera has a digital gimbal that allows to fulfill the assumption that the camera is always pointing to the ground. The vehicle was controlled through commands sent to it via Wi-Fi by a Matlab${}^{®}$ application running in a ground-based PC. The same ground-based application was used for capturing, via Wi-Fi, the sensor data from the drone. In this case, camera frames with a resolution of 856×480 pixels were captured at 24 fps. The altimeter signal was captured at 40 Hz. The range measurement between the UAV and the target was obtained by using the images and geometric information of the target. In experiments, the target was represented by a person walking with an orange ball over his head (See Figure 14). For evaluating the results obtained with the proposed method, the on-board GPS device mounted on the quadcopter was used to obtain a flight trajectory reference. It is important to note that, due to the absence of an accurate ground truth, the relevance of the experiment is two-fold: (i) to show that the proposed method can be practically implemented with commercial hardware; and (ii) to demonstrate that using only the main camera and the altimeter of Bebop 2, the proposed method can provide similar navigation capabilities than the original Bebop’s navigation system (which additionally integrate GPS, ultrasonic sensor, and optical flow sensor), in scenarios where a cooperative target is available.

Figure 14 shows a frame taken by the UAV on-board camera. The detection of the target is highlighted with a yellow bounding box. The search area of landmarks near the target is highlighted with a blue circle centered on the target. For the experiment, a radius of 1 m was chosen for the sphere centered on the target that is used for discriminating the landmarks. In this frame, some visual characteristics are detected in the image. The red cercles indicate those visual features that are not within the search area near the target, that is, inside the blue circle. Instead, the green circles indicate those detected features within the search area. The visual features that are found within the patch that corresponds to the target (yellow box) are neglected, this behaviour is to avoid considering any visual feature that belongs to the target as a static landmark of the environment.

Figure 15 shows both the UAV and the target estimated trajectories. This figure also shows the trajectory of the UAV given by the GPS and the altitude measurements supplied by the altimeter. Although the trajectory given by the GPS cannot be considered as a perfect ground-truth (especially for the altitude), it is still useful as a reference for evaluating the performance of the proposed visual-based SLAM method, and most especially if the proposed method is intended to be used in scenarios where the GPS is not available or reliable enough. According to the experiments with real data, it can be appreciated that the UAV trajectory has been estimated fairly well.

## 7. Conclusions

This work presented a cooperative visual-based SLAM system that allows an aerial robot following a cooperative target to estimate the states of the robot as well as the target in GPS-denied environments. This objective has been achieved using monocular measurements of the target and the landmarks, measurements of altitude of the UAV, and range measurements between UAV and target.

The observability property of the system was investigated by carrying out a nonlinear observability analysis. In this case, a contribution has been to show that the inclusion of altitude measurements improves the observability properties of the system. Furthermore, a novel technique to estimate the approximate depth of the new visual landmarks was proposed.

In addition to the proposed estimation system, a control scheme was proposed, allowing to control the flight formation of the UAV with respect to the cooperative target. The stability of control laws has been proven using the Lyapunov theory.

An extensive set of computer simulations and experiments with real data were performed to validate the theoretical findings. According to the simulations and experiments with real data results, the proposed system has shown a good performance to estimate the position of the UAV and the target. Moreover, with the proposed control laws, the proposed SLAM system shows a good closed-loop performance.

## Figures and Tables

**Figure 1 sensors-20-03531-f001:**
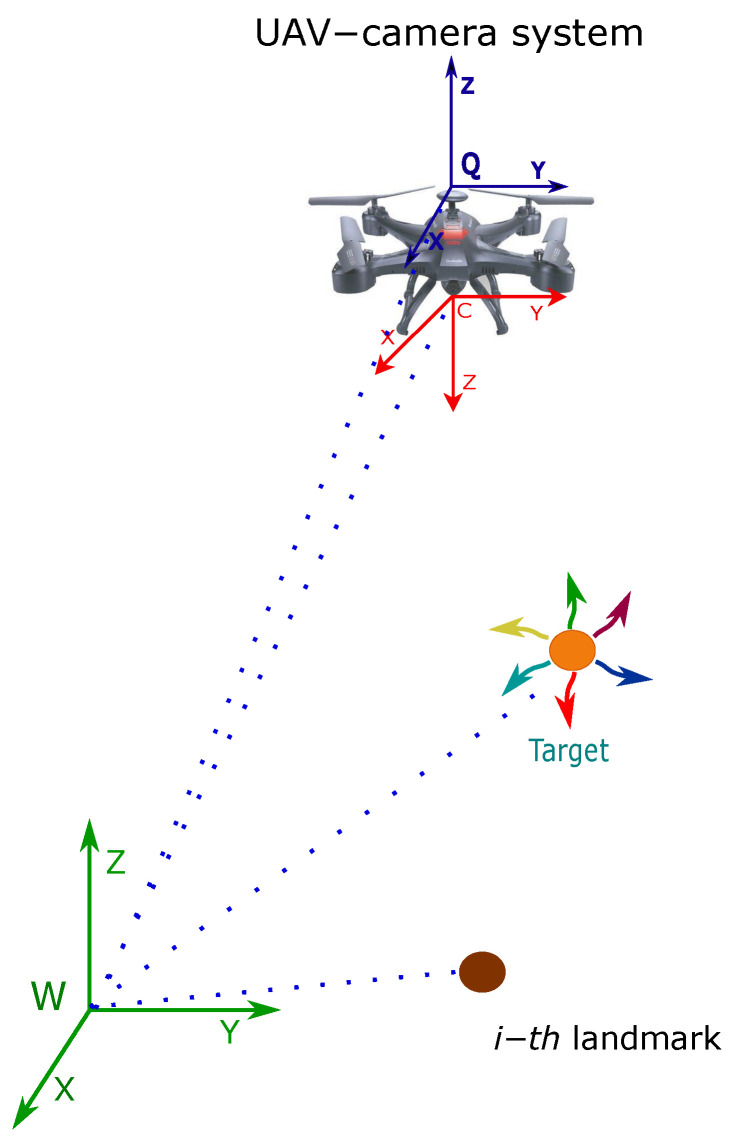
Coordinate reference systems.

**Figure 2 sensors-20-03531-f002:**
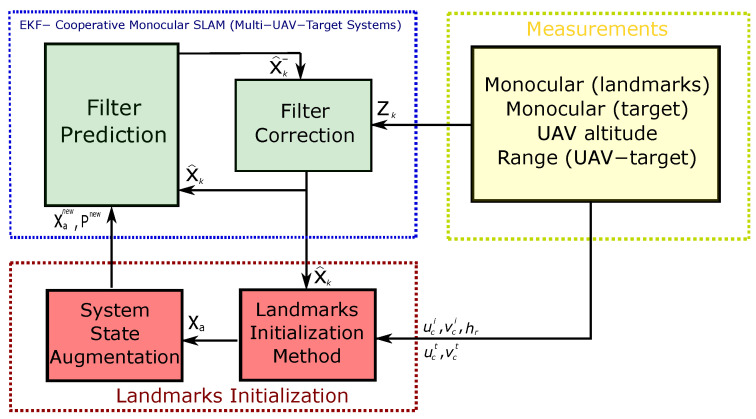
Block diagram showing the EKF-SLAM architecture of the proposed system.

**Figure 3 sensors-20-03531-f003:**
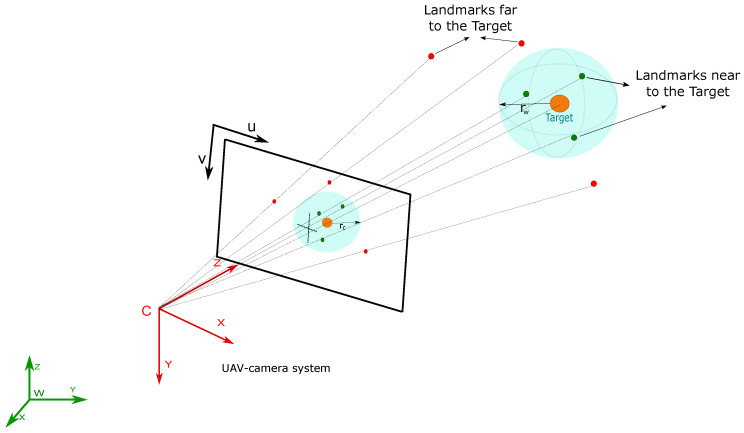
Landmarks selection method.

**Figure 4 sensors-20-03531-f004:**
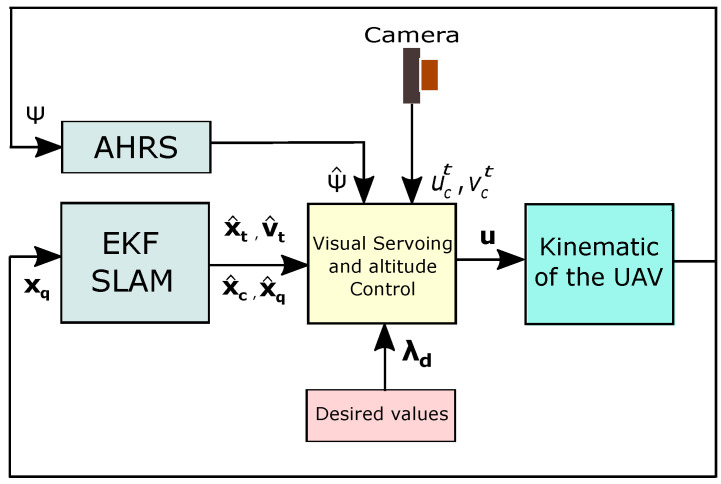
Control scheme.

**Figure 5 sensors-20-03531-f005:**
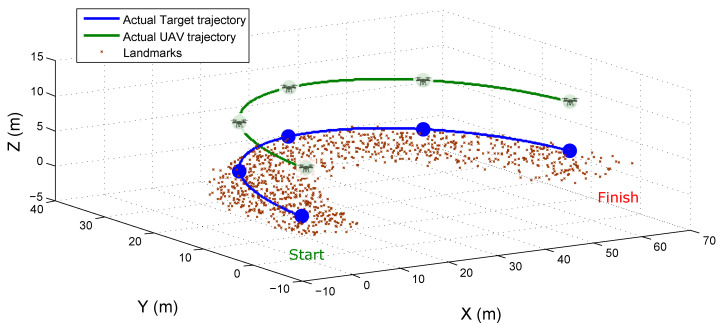
Actual UAV and target trajectories.

**Figure 6 sensors-20-03531-f006:**
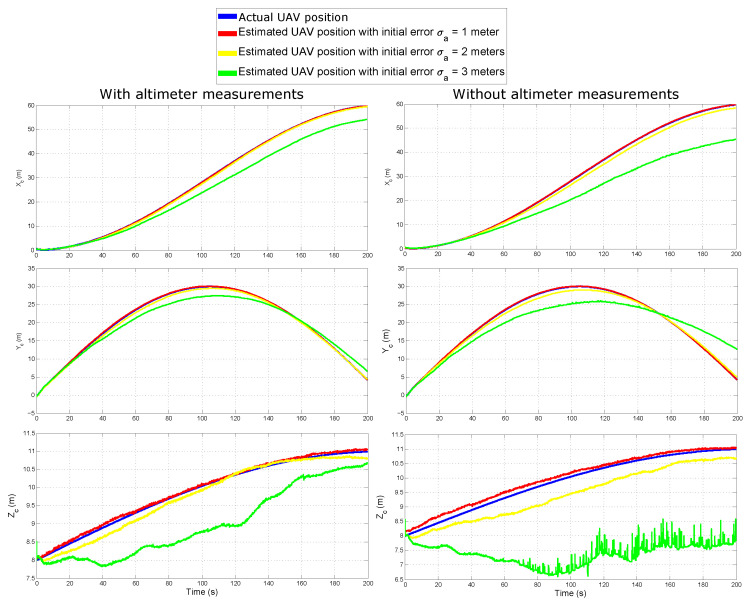
UAV estimated position.

**Figure 7 sensors-20-03531-f007:**
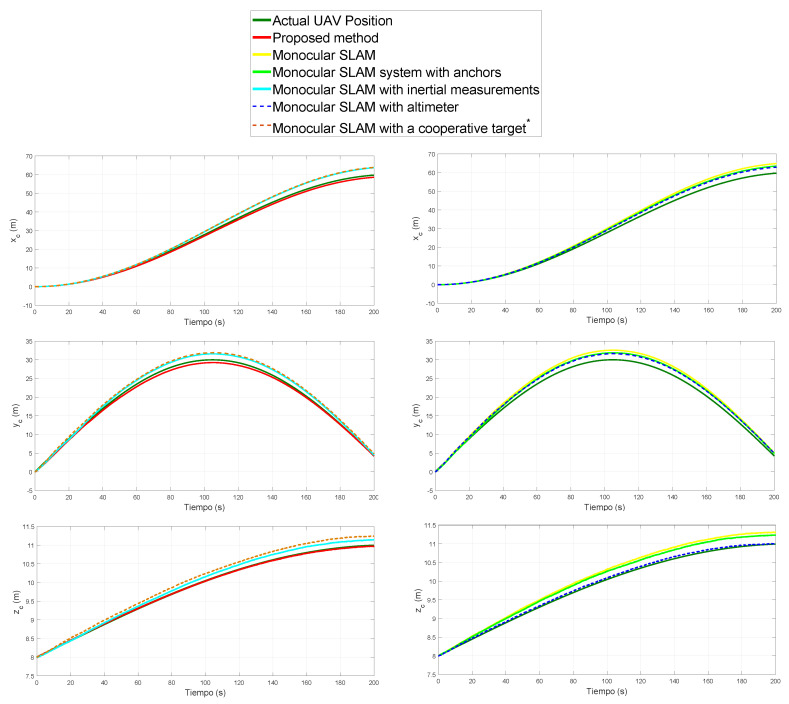
Comparison: UAV estimated position.

**Figure 8 sensors-20-03531-f008:**
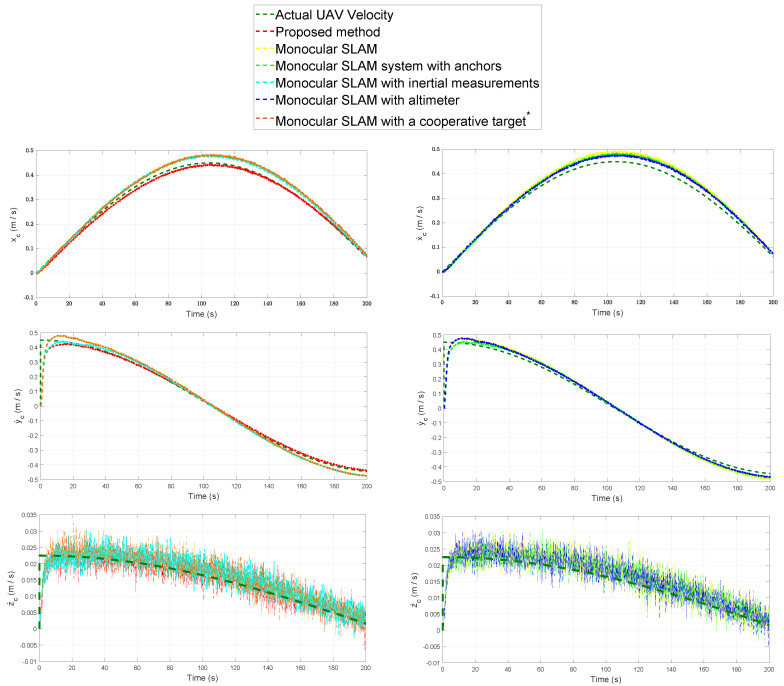
Comparison: UAV estimated velocity.

**Figure 9 sensors-20-03531-f009:**
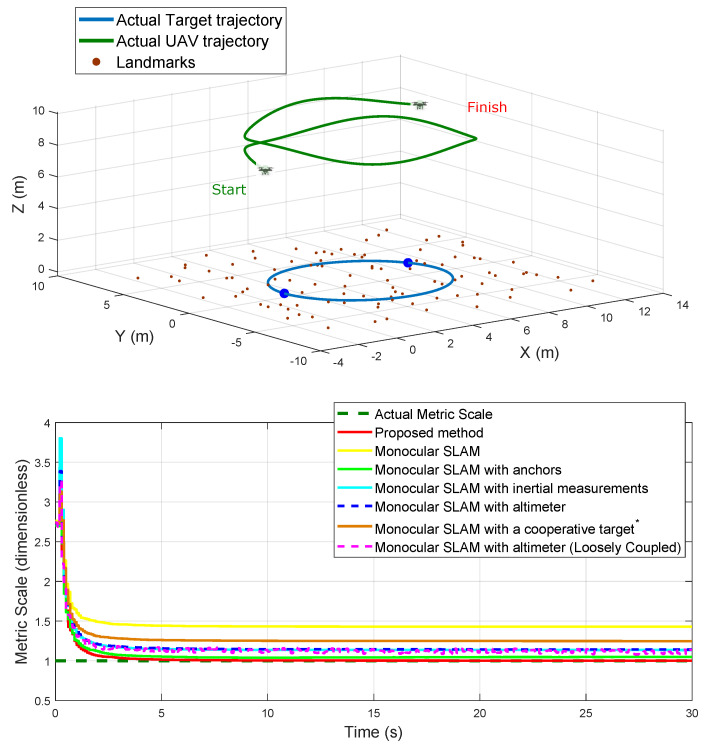
Case 1: Comparison of the estimated metric scale.

**Figure 10 sensors-20-03531-f010:**
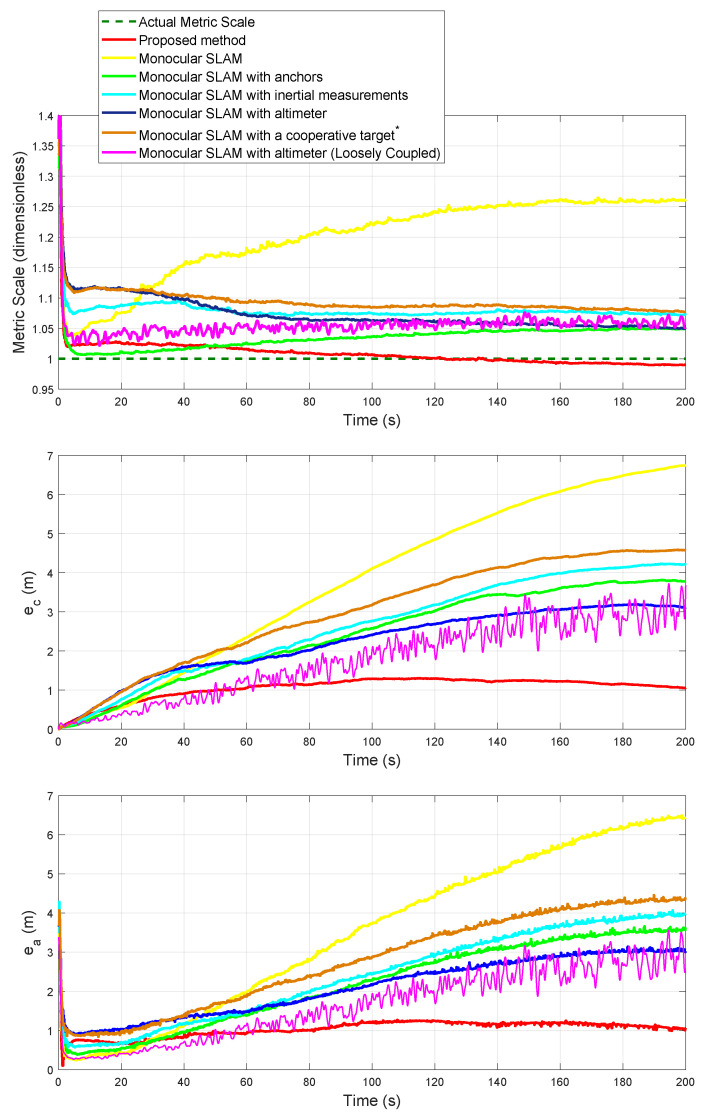
Case 2: Comparison of the estimated metric scale and Euclidean mean errors.

**Figure 11 sensors-20-03531-f011:**
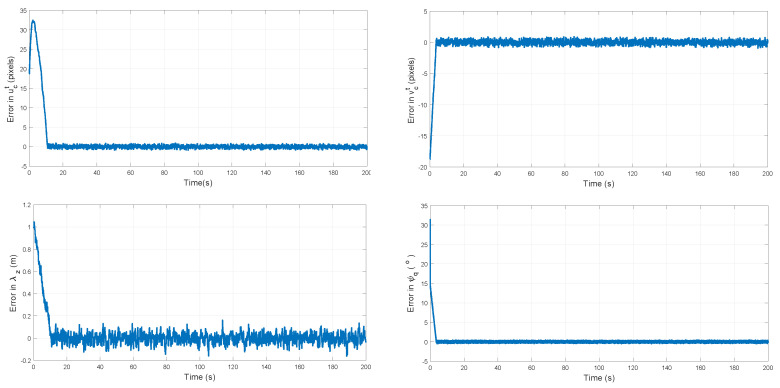
Errors with respct to $\lambda_{d}$.

**Figure 12 sensors-20-03531-f012:**
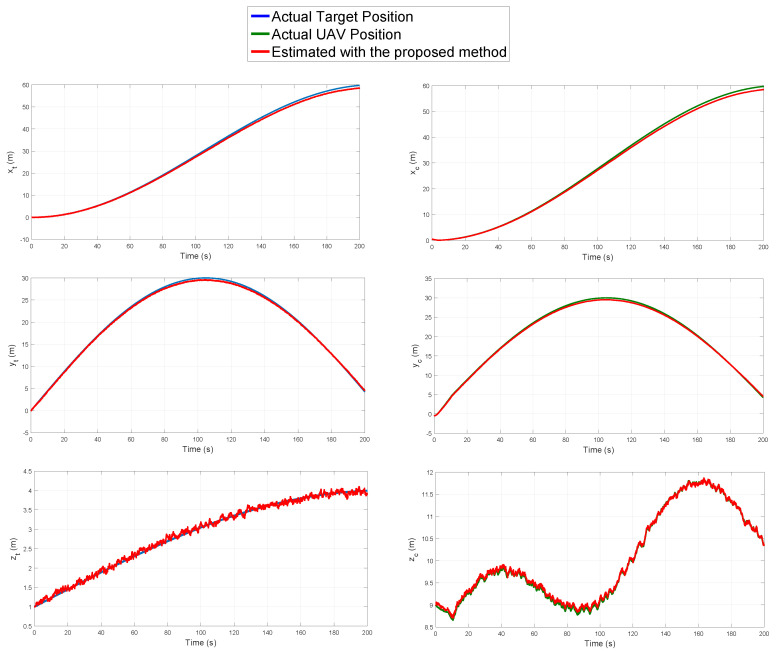
Estimated position of the target and the UAV obtained by the proposed method.

**Figure 13 sensors-20-03531-f013:**
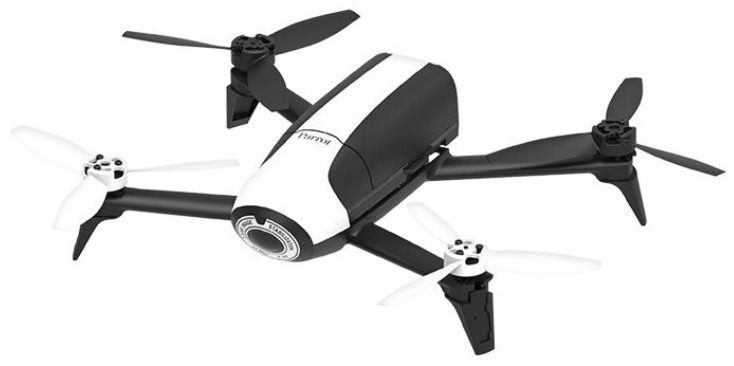
Parrot Bebop 2${}^{®}$ quadcopter.

**Figure 14 sensors-20-03531-f014:**
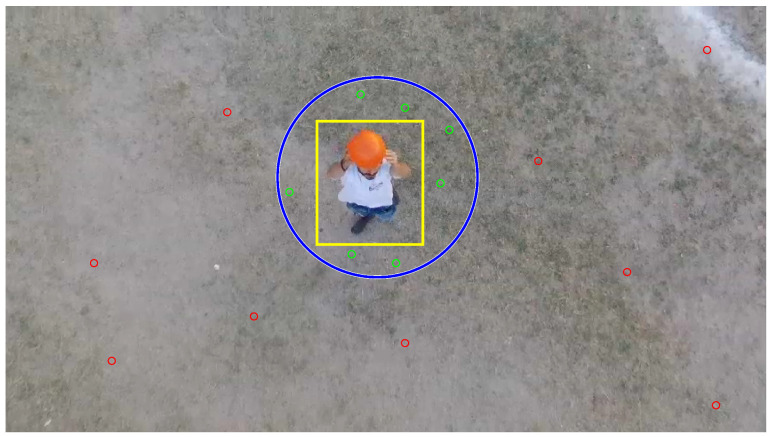
Frame captured by the UAV on-board camera.

**Figure 15 sensors-20-03531-f015:**
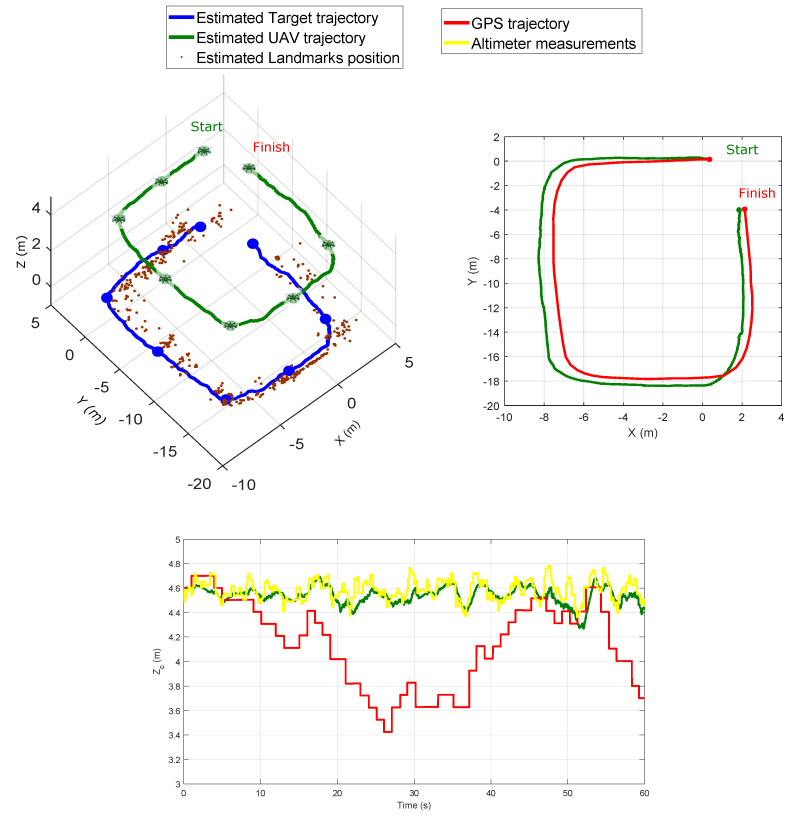
Comparison between the trajectory estimated with the proposed method, the GPS trajectory and the altitude measurements.

**Table 1 sensors-20-03531-t001:** Results of observability test.

	Unobservable	Unobservable	Observable
	Modes	States	States
Without altimeter measurements	4	x	-
With altimeter measurements	2	$x_{t}$, $y_{t}$$x_{c}$, $y_{c}$, $x_{a}^{i}$, $y_{a}^{i}$	$z_{t}$, $z_{c}$, $z_{a}^{i}$, $v_{t}$, $v_{c}$

**Table 2 sensors-20-03531-t002:** Mean Squared Error for the estimated position of the target ($MSEX_{t}$, $MSEY_{t}$, $MSEZ_{t}$) and the estimated position of the UAV ($MSEX_{c}$, $MSEY_{c}$, $MSEZ_{c}$).

		${\mathrm{MSEX}}_{t}\left(m\right)$	${\mathrm{MSEY}}_{t}\left(m\right)$	${\mathrm{MSEZ}}_{t}\left(m\right)$	${\mathrm{MSEX}}_{c}\left(m\right)$	${\mathrm{MSEY}}_{c}\left(m\right)$	${\mathrm{MSEZ}}_{c}\left(m\right)$
With Altimeter	$\sigma_{a}=1$ m	0.0075	0.0187	0.0042	0.0045	0.0151	0.0033
	$\sigma_{a}=2$ m	0.1214	0.2345	0.0302	0.1170	0.2309	0.0219
	$\sigma_{a}=3$ m	18.9603	3.0829	0.9351	18.9578	3.0790	0.8962
Without Altimeter	$\sigma_{a}=1$ m	0.0178	0.0139	0.0153	0.0145	0.0105	0.0207
	$\sigma_{a}=2$ m	1.7179	0.4689	0.2379	1.7078	0.4686	0.2084
	$\sigma_{a}=3$ m	80.9046	12.8259	7.3669	80.9000	12.8185	6.9981

**Table 3 sensors-20-03531-t003:** Total Mean Squared Error for the estimated position of the UAV.

Method	MSEX(m)	MSEY(m)	MSEZ(m)
Proposed method	0.5848	0.2984	0.0001
Monocular SLAM	9.1325	3.6424	0.0642
Monocular SLAM with anchors	4.9821	1.8945	0.0394
Monocular SLAM with inertial measurements	4.9544	1.2569	0.0129
Monocular SLAM with altimeter	3.5645	1.5885	0.0016
Monocular SLAM with a cooperative target	5.5552	1.9708	0.0367

**Table 4 sensors-20-03531-t004:** Total Mean Squared Error for the estimated position of the landmarks.

Method	MSEX(m)	MSEY(m)	MSEZ(m)
Proposed method	0.6031	0.2926	0.1677
Monocular SLAM	8.1864	2.8295	0.3861
Monocular SLAM with anchors	4.4931	1.4989	0.2701
Monocular SLAM with inertial measurements	4.4739	0.9979	0.3093
Monocular SLAM with altimeter	3.2397	1.2609	0.3444
Monocular SLAM with a cooperative target	5.0374	1.5394	0.3054

**Table 5 sensors-20-03531-t005:** Mean Squared Error for the estimated position of target, UAV and landmarks.

	MSEX(m)	MSEY(m)	MSEZ(m)
Target	0.5149	0.0970	0.0036
UAV	0.5132	0.0956	0.0018
Landmarks	0.5654	0.1573	0.2901

**Table 6 sensors-20-03531-t006:** Mean Squared Error for the the initial depth (MSEd) and position estimation of the landmarks.

	MSEd(m)	MSEX(m)	MSEY(m)	MSEZ(m)
Far from the target	13.4009	3.5962	2.5144	7.6276
Near to the target	1.6216	0.5188	0.1280	1.6482

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
