# Peer review of "Monocular Visual SLAM Based on a Cooperative UAV–Target System"

_sensors, 2020, doi:10.3390/s20123531_

Round 1

Reviewer 1 Report

In this paper the authors propose a monocular V-SLAM system with altitude measurements from a barometer and a vision-based distance measurement to a cooperative target. The landmarks are represented by both a Euclidean pose and inverse depth parameterization. The paper provides an observability analysis of the problem of the SLAM problem and a visual servo control law to follow the target. Finally, simulation and experiment data are provided. However, the strength of the contributions are weakened by the poor presentation and lack of rigour in the derivations.

  • The abstract, like much of the paper, can be improved by better defining the problem the paper is intending to solve. Phrases such as "the whole estimation problem" do not have any meaning. Furthermore, the actual state is only defined in the appendix (A2) and mentioned for the first time at the bottom of page 7. The literature review provides good references out of the vast literature on the subject, but downplays the success of other V-SLAM methods, and does not provide a strong motivation for why this exact setup is needed.
  • The paper repeatedly mentions the importance of recovering the metric scale, but does not define the scale, nor directly show its estimation.
  • The observability section is very difficult to read. At the very least it should start of by defining the state, the dynamics of the system, and the measurements. The majority of the variables are not defined. The observability in matrix O in (1) should be infinitely large. This is especially important when trying to show that something is not observable, and the higher order derivatives should only be dropped off with a proof showing that they do not increase the rank (also in line 147). The rank calculations of O should be expanded upon (even if left to an appendix). It is important to point out at this point (3.2) that x and thus the dim(x) has never been defined. Line 339 - "the right nullspace basis of O spans for N1" does not necessarily mean that all of the states are unobservable. The rank condition is only a sufficient condition of locally weakly observable. Table 1 - shows that the state is position and velocity of the uav and target, but O appears to include all of the landmarks.The observability study appear to conclude that the purpose of the work is measure the absolute position and velocity in the world, but this is unobservable. It would help the reader to point out that these values are not needed/important. The SLAM system is calculating the position and velocity within its own map. Also of importance is the relative distance between the target and the UAV, and this is partially measured. A stronger introduction/conclusion would really help the reader.
  • For the sensors : Stating that the barometer can directly measure the height of the UAV (or that a sensor that directly measures the height even exists) is an assumption that should be explained. Barometers are similar to GPS's in that they both don't work indoors. Barometers also provide relative measurements and not absolute measurements. This work also assumes a perfect non-drifiting AHRS would should be noted. Lastly, the distance to target measurement should explained along with its limitations. Line:363 Add a cite for the "standard" EKF-based SLAM scheme.
    Line 176 - On the first read of the paper this was the first time I realized that the acceleration is considered as 0 mean noise. This basically implies that the UAV and target have 0 (or constant) velocity. This is a large limitation that should be discussed and implies large drift during dynamic manueuvers.
  • 5 Target Tracking Control
    A kinematic model is a poor choice of models for a quadrotor due to the coupling of its attitude and linear acceleration, but would also be appropriate for mostly hover flight. Why does (42) have mixed reference systems? psi dot = omega is an assumption and should be stated as such.
  • Results
    There is no mention of velocity anywhere in the results, yet it is an important part of the state that is being estimated. Line 278: First mention of a gimbal and it's assumption. This is never explained. Line 288: If all of the initial conditions are exactly known then the initial positions of all of the landmarks in the camera should also be known exactly. Line 313: This needs to be more precise. The position is known exactly in the camera frame not the world frame. Line 318: This is an important point, but the theory and results in this paper do not show anything about scale. The converge properties of the scale is more nuanced then just motion along the vertical axis. In the comparative study is a little unfair to compare a strictly camera based method to a method that includes two other sensors. Other methods with altimeters and targets would be a better choice. Fig 8: Psi->There are some very large fast oscillations in the first second of flight. Is this physically possible or maybe dangerous for an actual quadrotor? Fig. 11: Z-> It appears to be more likely that the barometer was incorrect than the GPS. GPS problems are typically discrete jumps not slowly drifting up and down. It seems like this data shows a problem with the method (or at least the reliance on a barometer as an altimeter) rather than an advantage where GPS went wrong. Lastly, a large improvement to the paper would be a closed loop flight experiment using both the SLAM system and visual servoing.
  • A few minor issues:
    • the paper needs heavy editing of English/grammar
    • after a formulas sentences are continued with "where" starting indented.
    • equation numbers squished down
    • "T" should not always be capitalized for target. Only when referring to "the" specific target defined
    • Font in the figures is not consistent with the text
    • Fig. 1: Equation numbers might help the readability. Also the notation is not that helpful as most variables are X. Text is not vertically aligned
    • Fig. 3: Different fonts, ZOH is not necessary.
    • Acronym style is not consistent.
    • make sure all notation is defined immediately after first use.

Author Response

Please, find attached the responses in the file. 

Reviewer 2 Report

  1. It is difficult to understand the observability analysis section. The authors should present measurements first. 
  2. The observability analysis takes too much space. Is is possible to shorten the section as the results can be expected from physical sense.
  3. The unknown disturbance(Delta_g) in the stability analysis of Eq.(56) is missing.
  4. How about the matrix B^(-1) in Eq.(53). Does inverse always exist. Or should we use pseudo inverse then?
  5. I wonder where the set up using altimeter for SLAM is new in this study or has been done already in some previous studies.

Author Response

(The authors gave the same response as above.)

Reviewer 3 Report

This manuscript proposed an visual SLAM approach based on an UAV-target cooperative system in GPS-denied environments. Based on some strong constraints, such as monocular measurements of the Target and the landmarks, measurements of altitude of the UAV, range measurements between UAV and Target and AHRS information being available, the states of the UAV and the target can be estimated using monocular vision. A control law was also proposed to control the UAV following the cooperative target.

Although I do not think this research is valuable for real robotic applications because of those strong constraints, some theoretical results were obtained based on the observability analysis.

Please improve the figure 1 to make it clearer, such as those arrows in the block of EKF-UAV-Target SLAM.

Author Response

Please, find attached the responses in the file

Round 2

Reviewer 1 Report

I appreciate that the authors took my comments seriously and revised their methodology and presentation.
The new simulations demonstrate the performance of their system relative to other common approaches.
This comparison gives the paper more 'weight' in terms of contribution. The revisions in the literature
review also clarify the author's contribution with respect to the literature. A few points remain:

(Numbering from the previous letter)

(1),(9) The paper is easier much easier to read, however, the abstract still does not explicitly state what
problem it are trying to solve. 'The states of robot and target' needs to be explained. As previously mentioned
the 'state' could be anything. From the observability section it appears that the goal of the paper is to
estimate the velocities of the robot and target in a navigation frame. From the EKF-based SLAM section
is it appears that the goal of the paper is to estimate the position and velocity of the robot and target in
SLAM frame. Section 1.1 states the the objective is the estimation of the (unobservable) position of the
robot and target. It also states that the proposed method seeks to provide scale recovery (see 4) and drift
correction. Drift is never mentioned anywhere else in the paper.

(4) The scale is still not defined in the paper. It appears in the new Fig. 9 and is mysteriously set to
1. It is not inside of the observability analysis. More information on scale may be easier to find in
computer vision textbooks compared to SLAM literature. Another good reference is Weiss's PhD thesis
'Vision based navigation for micro helicopters'. In it he addresses the same method as proposed in this paper.

(6-8) 'The analytical proof required to show that the rank of the observability matrix remains constant
from some particular order derivative could be quite difficult'. Yes, this is why it would add value to the
paper. The abstract states 'extensive analysis of the system nonlinear observability', however, the robot
and the target's dynamics are linearized, scale is ignored, and a numerical rank that excluded the higher
order terms was used.

(22) The difference seen in the plot is within the error of GPS, but also within the error of the barometer.
Without having a more exact ground truth it is difficult to know. If the behop has a trust worthy altitude
measurement fused from different sensors perhaps it should be plotted too.

For the new figures it would help readability if a name was used instead of #1-5.
Why does method #2 have a time varying scale when it was assumed that the scale is perfectly known
from the anchors.

Author Response

Dear Reviewer, please find attached a letter wth the answers and comments to your valuable suggestions.

Round 3

Reviewer 1 Report

Observability analysis:
'it is important to note that even the majority of the more recent works in SLAM, still does not provide any kind of theoretical analysis or foundation for their methods'
Not including the theoretical analysis in a work is not the issue at hand. A large portion of this paper is dedicated to an observability analysis. So this analysis should be precise and rigorous

Scale:
The definition of scale is wrong. The scale of the map can be defined independent of the pose of the camera. But as mentioned it may sometimes be estimated by using the average of known depths of landmarks at particular point in time. It does not change given poor estimates of the pose of the camera. I would guess that the scale of the monocular slam system from fig. 9 is ~= 1.5 for 0<=t<=30s. If the other methods were to be loosely coupled and use the same monocular slam system on the same data then the scale should be the same for all of the methods. As the proposed method is tightly coupled and does not include the scale (it seems to be taken as measurement noise) the predicted map will be somewhere between the scaled map and actual values depending on a lot of different factors.

Drift:
In fig 9 only an error in the ratio of estimated depth to depth can be seen.
In the figure in the response letter there is not enough information to know if there is a scale drift or a scale estimate error. The convergence (or divergence) of the estimates is not drift. Dead reckoning error has absolutely nothing to do with this problem (there is no IMU in the purely monocular SLAM nor the proposed method). The pose drift is from the incremental estimation error in the camera pose/its covariance. The scale drift is the incremental estimation of the map between landmarks. The two values should not be confused. Re-time-varying. Yes, the scale is not constant, but it drifts very slowly and will almost not be seen in 30s, but of course this depends on a lot of different factors.

Weiss:
I agree that Weiss's method (loosely coupled, optimization based slam + filter based fusion) versus the proposed method (tightly coupled, filter based) are quite different. His thesis was just an example of something similar that includes an observability analysis and scale. The proposed method has better results, but also include stronger assumptions.

(6-8),(22) Looks good.

Another proofread for English is still required. eg. the new figure has the x axis labeled in Spanish.

Author Response

Dear reviewer #1, please find attached a letter with response to your question. 

This manuscript is a resubmission of an earlier submission. The following is a list of the peer review reports and author responses from that submission.